# Serum amyloid A promotes glycolysis of neutrophils during PD-1 blockade resistance in hepatocellular carcinoma

Meng He [1,4], Yongxiang Liu[1,4], Song Chen [1,2,4], Haijing Deng[3,4], Cheng Feng[1,4], Shuang Qiao [1], Qifeng Chen[1,2], Yue Hu[1,2], Huiming Chen[1], Xun Wang[1], Xiongying Jiang[1], Xiaojun Xia [1], Ming Zhao [1,2] ✉ & Ning Lyu [1,2] ✉

The response to programmed death-1 (PD-1) blockade varies in hepatocellular carcinoma (HCC). We utilize a panel of 16 serum factors to show that a circulating level of serum amyloid A (SAA) > 20.0 mg/L has the highest accuracy in predicting anti-PD-1 resistance in HCC. Further experiments show a correlation between peritumoral SAA expression and circulating SAA levels in patients with progressive disease after PD-1 inhibition. In vitro experiments demonstrate that SAA induces neutrophils to express PD-L1 through glycolytic activation via an LDHA/STAT3 pathway and to release oncostatin M, thereby attenuating cytotoxic T cell function. In vivo, genetic or pharmacological inhibition of STAT3 or SAA eliminates neutrophil-mediated immunosuppression and enhances antitumor efficacy of anti-PD-1 treatment. This study indicates that SAA may be a critical inflammatory cytokine implicated in anti-PD-1 resistance in HCC. Targeting SAA-induced PD-L1$^+$ neutrophils through STAT3 or SAA inhibition may present a potential approach for overcoming anti-PD1 resistance.

Hepatocellular carcinoma (HCC) is the third leading cause of cancer-related mortality[1]. Most patients are diagnosed at an advanced stage and have a poor prognosis due to the palliative nature of systemic treatments[2]. Immunotherapies that block programmed death ligand-1 (PD-L1) checkpoints in combination with anti-angiogenesis or other checkpoints, such as cytotoxic T-lymphocyte-associated antigen 4 (CTLA-4), are currently recommended as first-line treatment for advanced HCC (aHCC)[2,3]. On the other hand, the risks of liver function disruption and gastrointestinal hemorrhage are widespread as deadly blows for patients with aHCC. These risks are typically employed as selection criteria in pivotal clinical trials for systemic combination therapies[4,5]. Anti-PD-1 monotherapy has been demonstrated with a well-tolerated safety profile and moderate antitumor activity for aHCC[6,7]. Nivolumab, an immunoglobulin (IG)

G4 monoclonal antibody to PD-1, is the only approved systemic medication for aHCC patients with a Child-Pugh Score of 8 (Class B)[8]. Hence, identification of biomarkers to classify patients for PD-1 blockade and investigation of resistance mechanisms in HCC immunotherapy have become increasingly important in recent years[9].

Peripheral inflammation-related biomarkers intuitively reflect the systematic immune status closely related to the tumor micro-environment (TME) and PD-1 antibody response[10–12]. In CheckMate 040 clinical trials, biomarker analysis in HCC revealed various inflammatory genes and cells indicating the neutrophil to lymphocyte ratio (NLR) and platelet-to-lymphocyte ratio (PLR) that were related to overall survival (OS) following the course of nivolumab[10]. In addition, a multicenter retrospective study created an externally

[1]State Key Laboratory of Oncology in South China, Sun Yat-sen University Cancer Center, Guangzhou, Guangdong, China. [2]Department of Minimally Invasive Interventional Therapy, Sun Yat-sen University Cancer Center, Guangzhou, Guangdong, China. [3]Department of Pathology, The University of Hong Kong, Hong Kong, China. [4]These authors contributed equally: Meng He, Yongxiang Liu, Song Chen, Haijing Deng, Cheng Feng. ✉e-mail: zhaoming@sysucc.org.cn; lvning@sysucc.org.cn

validated score (i.e., CRAFTY) based on baseline levels of C-reactive protein (CRP) and alpha-fetoprotein (AFP) to predict response to PD-1/PD-L1-based immunotherapy[12]. These studies indicate that a thorough examination of inflammatory cytokines is essential for understanding the determinants of PD-1 antibody responses in HCC.

Serum amyloid A (SAA) 1 and 2 are acute inflammatory proteins induced during sepsis[13], and SAA is prominently derived from the liver[14]. It is the signal transducer and activator of transcription 3 (STAT3) in hepatocytes that promote the subsequent production of SAA[15]. SAA is involved in various immunological activities, including effects on cytokine synthesis and neutrophil chemotaxis to mediate immune escape[16]. Furthermore, SAA can act as a biomarker for the resistance of PD-1 antibodies in advanced non-small cell lung cancer[17]. However, the potential mechanisms underlying the participatory or regulatory role of SAA in the immune regulation of the TME remains unexplored.

Neutrophils, one of the target cells of SAA, are extensively involved in regulating tumor initiation, progression, metastasis, and immunotherapy response[18]. Tumor-associated neutrophils have been reported to suppress T-cell function in a PD-L1-dependent fashion and contribute to tumor progression in gastric cancer[19]. Additionally, PD-L1+ neutrophils can mediate the resistance to T-cell-dependent anti-tumor immunity of lenvatinib in HCC[20]. Therefore, investigation on regulation of neutrophils may help to identify potential targets for immune-based anticancer therapies for HCC[21,22]. Furthermore, the metabolic plasticity of neutrophils contributes to their functional diversity[23]. It is well known that neutrophils have a high capacity for glycolysis, and studies have validated the upregulation of glycolysis in tumor-associated neutrophils[24]. However, it has not been determined if targeting neutrophil metabolism is synergistic with cancer immunotherapy or if SAA contributes to the control of neutrophils in human HCC tumor microenvironments and mediates resistance to PD-1 antibodies.

In this work, we aim to identify circulating inflammatory biomarkers associated with anti-PD1 antibody response. We determine that SAA is an essential inflammatory cytokine implicated in anti-PD-1 tolerance in HCC. Herein, hepatocyte-derived SAA may mechanistically recruit neutrophils into the TME and stimulate PD-L1 expression on neutrophils via promoting glycolysis. The SAA-related PD-L1+ neutrophils may facilitate the immune escape by inhibiting T cell cytotoxicity. Our findings primarily demonstrate a liver-specific mechanism contributing to immunotherapy resistance. Our study identifies a peripheral biomarker to predict and monitor the efficacy of PD-1 blockade and further provides a new potential combination therapy against HCC.

## Results

### SAA correlates with the efficacy of PD-1 blockade in aHCC patients

We evaluated the relationship between patients' clinical outcomes and the median blood level of candidate circulating biomarkers, including 16 markers in 52 HCC patients undergoing PD-1 blockade monotherapy (Supplementary Fig. 1a and Supplementary Table 1). The 16-biomarker panels were divided into three categories of inflammatory markers (SAA, CRP, NLR, PLR, and LDH), TH1/TH2 cytokines (IL-2, IL-4, IL-5, IL-6, IL-8, IL-10, IFNγ, TNF, and GM-CSF), and HCC tumor markers (AFP and PIVKA-II). The log-rank test analysis of the Kaplan–Meier survival curve showed a high level of inflammatory markers of SAA (hazard ratio [HR] 3.63; 95% confidence interval [CI], 1.46–8.98; $P = 0.008$) and statistically significant PLR (HR 3.63; 95% CI, 1.47–8.99; $P = 0.008$) associated with poorer median OS (Supplementary Fig. 2a). For progression-free survival (PFS), SAA (HR 2.54; 95% CI, 1.27–5.06; $P = 0.006$) and tumor marker-PIVKA-II (HR 2.23; 95% CI, 1.13–4.42; $P = 0.018$) were potential prognostic factors (Supplementary Fig. 2b). For response, all

predicting factors (PD vs. NPD) were inflammatory markers, including SAA ($P < 0.001$) and CRP ($P < 0.01$) (Supplementary Fig. 2c). The LASSO multivariate analysis further demonstrated that SAA, CRP and NLR were predicting factors for tumor response (Supplementary Fig. 2d). The baseline circulating SAA level showed a higher area under the curve (AUC) value for predicting PD status than other selected markers (AUC: 0.848; 95% CI, 0.83–0.92; $P < 0.0001$) (Fig. 1a). Then, our investigation revealed that the cut-off value (20.0 mg/L) of circulating SAA levels, as defined by ROC analysis of tumor response, may serve as a more accuracy for prognosing factor for survival outcomes of aHCC patients undergoing PD-1 blockade monotherapy compared to utilizing the median level (12.3 mg/L) of SAA (OS: HR 4.44; 95% CI, 1.70–11.57; $P = 0.001$; PFS: HR 2.93; 95% CI, 1.38–6.21; $P = 0.001$; Supplementary Fig. 2e, f and Supplementary Table 2).

From the above 52 patients, we obtained both tumor and peritumor tissues from six patients before the treatment by core needle biopsy and processed for RNA-seq (NPD: $n = 3$; PD: $n = 3$) (Supplementary Fig. 1b and Supplementary Table 3). The principal component analysis (PCA) of RNA-seq revealed that the transcriptomic profile of peritumor tissues was distinguished between the NPD and PD patients, while that of tumor tissues could not be differentiated (Supplementary Fig. 3a). Gene ontology (GO) enrichment analysis found that the acute inflammatory response pathway is markedly upregulated in the peritumor tissue of PD patients (Supplementary Fig. 3b). Differentially expressed genes (DEG) analysis further showed that the upregulation of *SAA1* was higher than other acute-inflammatory proteins such as *CRP* (Supplementary Fig. 3c). Moreover, the expression of cytoplasmic SAA in peritumor was also higher in PD samples (NPD: $n = 5$; PD: $n = 4$) (Fig. 1b, c). Further, the circulating SAA level was positively correlated with local peritumoral SAA protein expression by using Pearson correlation analysis in a large population ($r = 0.558$; $P < 0.001$) (Fig. 1d). Given the above information, SAA in both circulation and peritumoral areas might have a parallel effect on predicting tumor responses and could potentially serve as prognostic factors for survival outcomes in aHCC patients treated with anti-PD-1 monotherapy.

Besides, in the population treated with anti-PD-1 plus tyrosine kinase inhibitors (TKIs), as locoregional chemotherapy, or both, a high baseline circulating SAA level (the cut-off value of 20.0 mg/L) was also significantly associated with poor clinical outcomes in aHCC (Supplementary Fig. 4a–e and Supplementary Table 4). The clinical outcome of three typical cases indicated that dynamic changes in circulating SAA levels can timely reflect radiological response in aHCC patients receiving anti-PD-1 immunotherapy (Fig. 1e–g). For further confirmation, the changes in circulating SAA levels during the treatment were tested in 17 patients with aHCC who received anti-PD-1-based therapy. In both NPD and PD patients, the median circulating SAA level was remarkably increased upon disease progression (Fig. 1h and Supplementary Fig. 5). Besides, the additional validation cohort ($n = 138$) demonstrated a significant association between SAA levels (using a cut-off value of 20.0 mg/L) and tumor responses as well as survival outcomes in aHCC patients treated with anti-PD-1-based treatment (Supplementary Fig. 4f, g and Supplementary Table 5).

### Peritumoral SAA-related PD-L1+ neutrophils affect anti-PD-1 response

By comparing the mRNA levels of *SAA* in paired tumoral and peritumoral tissues from the TCGA database, we found that *SAA* was expressed at a considerably higher level in peritumoral tissues than in tumor tissues, demonstrated by multiplexed immunohistochemistry (mIHC) analysis in paired HCC tissue microarray (TMA) and immunohistochemistry (IHC) analysis of both resectable HCC specimens and mouse tissues (Supplementary Fig. 6 and Supplementary Table 6). Further, we found that peritumoral SAA production was mainly dependent on the activation of STAT3 signaling in hepatocytes (Supplementary Fig. 7).

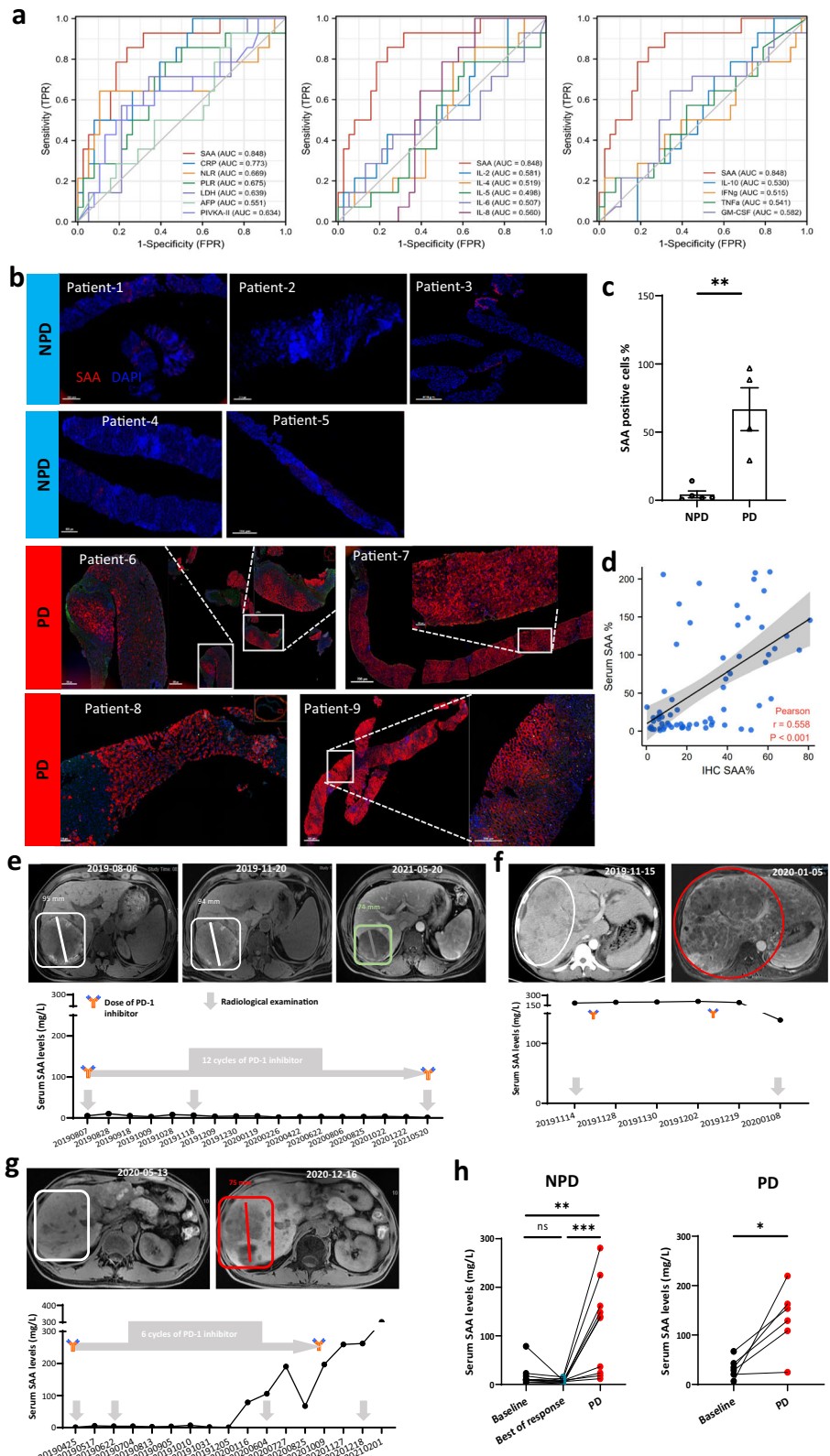

In the TCGA database, we assessed the relationship between SAA and tumor-infiltrating immune cells in HCC TME and found that neutrophils had the highest positive association with SAA (r = 0.47, $P < 0.0001$) (Supplementary Fig. 8a, b). By comparing the co-expression of SAA and infiltrating immune cells between high and low SAA HCC peritumor TMA in local samples ($n = 160$), neutrophils (CD15+) ($P < 0.0001$), and macrophages (CD68+) ($P < 0.01$) were

significantly increased, while CD8+ T cells were significantly decreased in the SAA-high cohort (Fig. 2a and Supplementary Fig. 8c). Furthermore, PD-L1+ neutrophils, but not PD-L1+ macrophages, had a positive relationship with peritumoral SAA expression ($P < 0.01$) (Fig. 2b). In resected tissues from NPD ($n = 14$) and PD ($n = 5$) patients who received anti-PD-1 immunotherapy (Fig. 2c and Supplementary Fig. 8d), both the protein expression levels of SAA and the quantity of PD-L1+

**Fig. 1 | SAA is associated with outcomes in aHCC patients treated with anti-PD-1. a** Fifteen biomarkers are selected for differentiating PD and NPD patients treated with the anti-PD-1 monotherapy using the ROC curves and the corresponding AUC values. The data reveals that baseline blood SAA had the highest value (AUC: 0.848) in predicting tumor response under the RECIST (v.1.1) criteria. $n = 52$ patients. **b** The pre-therapeutic cytoplasmic expression of SAA is significantly higher in PD patients ($n = 4$) than NPD patients ($n = 5$) using the mIHC analysis. **c** The histogram shows that the percentage of SAA-positive cells in PD patients' tissue samples is remarkably elevated than in NPD patients. **d** SAA expression levels in matched circulating and tissue samples from aHCC patients ($n = 103$) by using the Latex Enhanced Immunoturbidimetric Method and IHC, respectively. The Scatter plots show a positive correlation between the levels of peripheral blood SAA and peritumoral SAA expression. **e–h** Dynamic monitoring of blood SAA indicated that SAA could be a potential biomarker for predicting primary and secondary resistance to PD-1

inhibitors. **e** A patient with stably low SAA (<20.0 mg/L) has a durable tumor response and favorable clinical outcome of 24 months of progression-free survival. **f** A patient with pre-treatment high SAA shows disease progression after the anti-PD-1 therapy. **g** The increase in circulating SAA level is parallel to the progression of HCC in a patient with secondary resistance to the anti-PD-1 inhibitor. **h** In patients with secondary resistance to PD-1 blockade ($n = 10$ patients), the blood SAA level is significantly higher than both baseline SAA, and SAA detected at tumor response; in patients with primary resistance ($n = 6$ patients), there is also a difference between baseline SAA and SAA at tumor assessment. Statistical data presented in this figure show mean ± SEM. ns indicates $P > 0.05$, $*P < 0.05$, $**P < 0.01$ and $***P < 0.001$, by two-sided Student's $t$ test (**c** and right panel of **h**), two-sided Pearson correlation analysis (**d**), or one-way ANOVA (left panel of **h**). Source data and exact $P$ values are provided as a Source Data file.

neutrophils were significantly elevated in PD cases compared to those in NPD cases (Fig. 2d and Supplementary Fig. 9). Cellular spatial relationship map analysis showed that SAA+ hepatocytes spatially correlated with neutrophils (Supplementary Fig. 8e, f) and PD-L1+ neutrophils (Fig. 2e and Supplementary Fig. 9) in the peritumoral region of HCC tissues.

## SAA induces PD-L1 upregulation of neutrophils via STAT3 pathway

In our study, SAA had an independent effect on neutrophils in the blood samples collected from patients with aHCC (Supplementary Fig. 10a and Supplementary Table 7). The cell viability of neutrophils at 24 h was higher in the SAA-treated group than that in the untreated group (Supplementary Fig. 10b). In addition, the apoptosis of neutrophils was found to be delayed in the SAA-treated group by annexin V and propidium iodide detection (Supplemental Fig. 10c). Consistent with the correlation between SAA and PD-L1+ neutrophil in vivo, we found that SAA protein could induce PD-L1 expression on neutrophils in a dose and time-dependent manner in vitro (Fig. 3a, b).

A recent study revealed that activated STAT3 could induce PD-L1 expression of neutrophils in the microenvironment of gastric cancer[19]. In HCC, the upregulation of p-STAT3 was also observed in SAA-induced PD-L1+ neutrophils (Fig. 3c). Besides, the PD-L1 expression of neutrophils can be elevated by SAA protein (1 μg/mL) in a time-dependent manner via the p-STAT3 (Fig. 3d). Meanwhile, the glyceraldehyde-3-phosphate dehydrogenase (GAPDH), a rate-limiting enzyme that regulates aerobic glycolysis, was increased in SAA-treated neutrophils in a time-dependent manner (Fig. 3d). In the five-plex IHC image, PD-L1+ neutrophils exhibited positive staining of p-STAT3 and located close to SAA+ hepatocytes in the HCC peritumoral area (Fig. 3e). Finally, STAT3 inhibitor (napabucasin/BBI608) could shorten the lifespan of neutrophils (Supplementary Fig. 10b) and decrease the percentage of PD-L1+ neutrophils induced by SAA treatment (Fig. 3f).

## SAA induces PD-L1 expression through glycolytic activation

To further explore the effect of SAA stimulation on neutrophils, we performed RNA-seq of SAA-treated neutrophils at different time points (0, 30 min, 6 h, and 12 h after SAA stimulation) (Fig. 4a, b and Supplementary Fig. 11a). GO term analysis of the transcriptomic profiles showed that the cellular metabolic process (gene number: 4492, Q value: 58.72) was activated in SAA-treated neutrophils (Fig. 4c). As previously shown in Fig. 3, SAA induced STAT3 activation and increased GAPDH expression in PD-L1+ neutrophils. Therefore, we assumed that SAA might induce PD-L1 expression on neutrophils through glycolytic activation in HCC. In line with this, we first identified increased gene expression in terms of neutrophil activation and dysregulation (*FGR, AKR1A1, ANXA2, and METTL6*), glycolysis (*PGM3, PKM, ENO3, ALDOC, LDHB and PFKL*), and *PD-L1* (Fig. 4d and Supplementary Fig. 11b). Next, the upregulation of glucose uptake and lactate secretion, as well as the increase of fluorescent 2-NBD-glucose

consumption, were observed in SAA-treated neutrophils (Fig. 4e and Supplementary Fig. 11c). Moreover, the Glut1 expression on SAA-exposed neutrophils rapidly elevated within 6 h (Supplementary Fig. 11d). Seahorse analysis of extracellular acidification rate (ECAR) validated the activation of glycolysis in SAA-associated neutrophils (Supplementary Fig. 11e). Moreover, despite SAA incubation, PD-L1 can be downregulated by glycolysis inhibitor 2-Deoxy-D-glucose (2DG) in neutrophils derived from the blood of patients with HCC (Fig. 4f).

To further identify the specific glycolytic regulator to induce PD-L1 expression, we first observed that transcription of glycolytic enzyme genes, including *GLUT1, PKM2, LDHA, GAPDH, PFKL*, and *ALDOC*, increased in SAA-treated neutrophils derived from the peripheral blood of HCC patients (Fig. 4g). Likewise, several glycolytic enzyme genes, including *PKM2, PFKFB3, LDHA, GADPH, PFKL, ALDOA, ALDOC*, and *ALDOC*, were also upregulated in SAA-treated neutrophils derived from healthy donors (Supplementary Fig. 11f). We then found that protein expression of PKM2, LDHA, and monocarboxylate transporter 4 (MCT4, a high-affinity lactate efflux transporter) were markedly elevated in neutrophils after SAA expose (Supplementary Fig. 12a). The previous study reported that hypoxia-inducible factor 1α (HIF-1α) could activate PKM2 to participate in glucose metabolism reprogramming in cancer cells[25]. In addition, PKM2 could promote pro-inflammatory gene activation of neutrophils by STAT3 phosphorylation[26,27]. However, neither a PKM2 nuclear translocation antagonizer (TEPP) nor a HIF-1α inhibitor (α-ketoglutarate) could affect the expression of PD-L1 in SAA-stimulated neutrophils derived from the blood of HCC patients (Supplementary Fig. 12b, c).

To determine whether LDHA mediated the upregulation of PD-L1 expression in neutrophils, we tested the correlation between LDHA and STAT3 in LIHC in the TIMER 2 database and found that the expression of LDHA was positively correlated with STAT3 ($r = 0.488$, $P < 0.001$) (Supplementary Fig. 12d). We then inhibited LDHA by using FX-11 (an LDHA inhibitor) in SAA-incubated neutrophils and found that FX-11 could significantly abrogate the stimulated effect of PD-L1 expression in SAA-treated neutrophils (Fig. 4h, i). Besides, in the six-plex IHC images, p-STAT3+PD-L1+ neutrophils exhibited positive staining of LDHA and located close to CD8+ T cells in the HCC peritumoral area (Supplementary Fig. 12e).

## SAA-treated neutrophils increase OSM to inhibit CD8+ T cells activity

In our study, inflammatory response-related genes (e.g., *OSM, IL-1B*, and *IL-6*) were activated as early as 30 min following SAA stimulation in neutrophils (Fig. 5a). Additionally, the mRNA and protein expression of critical inflammatory response genes of N2 phenotype neutrophils, including IL-1β, ARG1, IL-10, and OSM, were also increased after SAA stimulation by qPCR and western blotting analysis, respectively (Fig. 5b-c). In SAA-treated neutrophils, the production of OSM, but not ARG1, was significantly upregulated (Fig. 5d). mIHC images showed that after being double-stained with an anti-MPO antibody

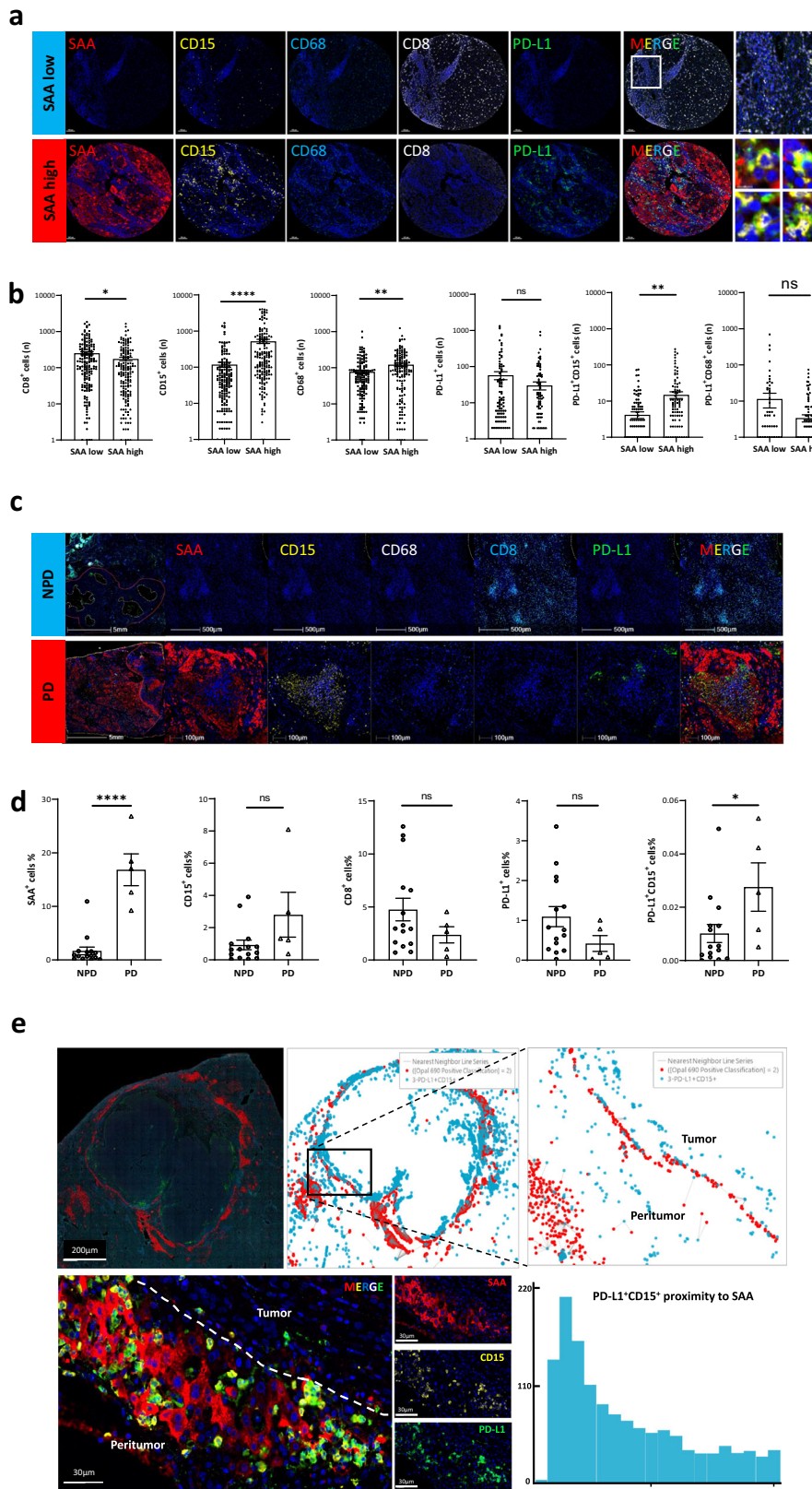

**Fig. 2 | Association between local SAA and immune cell infiltration in HCC.**
**a**, **b** mIHC images show the simultaneous detection of DAPI (blue), SAA (red), CD15+ neutrophils (yellow), CD68+ macrophages (brilliant blue), CD8+ T cells (white), and PD-L1 (green) in human HCC peritumor TMA. The count of infiltrating immune cells between SAA high and SAA low samples is compared using a two-sided Student's $t$ test. $n = 160$ patients. **c**, **d** In resected HCC specimens, mIHC images show the co-expression of local SAA (red), CD8+ T cells (brilliant blue), CD15+ (yellow), and CD68+ (white). Both the expression of SAA and PD-L1+CD15+ neutrophils are higher in PD ($n = 5$ patients) cases than in NPD ($n = 14$ patients) cases by a two-sided Student's $t$ test. **e** The cell spatial diagram of HALO analysis shows that the distance between most PD-L1 neutrophils and SAA+ cells was within the range of 5–50 μm. Statistical data presented in this figure show mean ± SEM. ns indicates $P > 0.05$, *$P < 0.05$, **$P < 0.01$ and ****$P < 0.0001$. Source data and exact $P$ values are provided as a Source Data file.

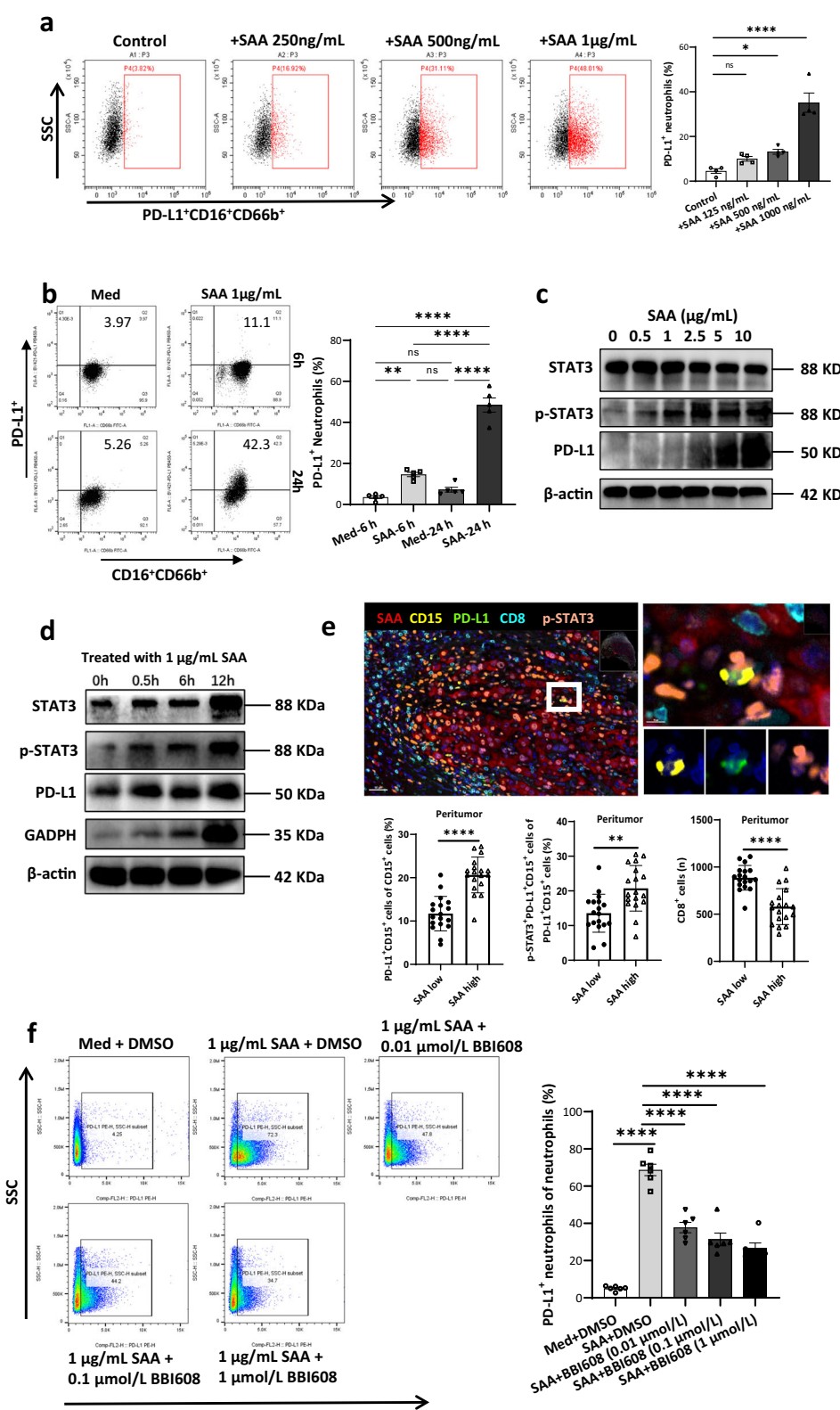

(a neutrophil marker) and an anti-OSM antibody, OSM+MPO+ cells were predominantly located in the invasion margin of HCC (Fig. 5e). Furthermore, it was observed that SAA-treated neutrophils exerted a suppressive effect on the production of interferon-γ (IFN-γ) and tumor necrosis factor (TNF) in autologous CD8+ T lymphocytes. However, this suppressive effect was effectively abrogated by the addition of an OSM inhibitor (OSMi, OSM-SMI-10B) (Fig. 5f). These findings

suggested that OSM is involved in mediating the inhibitory effects of SAA-treated neutrophils on CD8+ T lymphocytes.

## SAA blockade or *Saa1* ablation enhances anti-PD-1 efficacy in vivo

We next tested the effect of SAA blockade on anti-PD1 efficacy on HCC mouse models. C57B/6 J mice implanted with Hep1-6-luci+

**Fig. 3 | The relationship of local SAA, nuclear p-STAT3, and PD-L1⁺ neutrophils.** a FACS analysis shows the PD-L1 expression on neutrophils exposed to different concentrations of SAA (0, 250, 500, and 1000 ng/mL) for 24 h. Statistical analysis of PD-L1 expression on neutrophils indicates that SAA induced PD-L1 expression on neutrophils in a concentration-dependent manner. $n = 4$ biologically independent samples. b Flow cytometry images show that PD-L1 is rapidly upregulated after neutrophils are exposed to SAA (1000 ng/mL) for 6 h. $n = 5$ biologically independent samples. c Western blots show the protein expression of PD-L1, STAT3, and p-STAT3 in neutrophils after being exposed to different concentrations of SAA (0, 0.5, 1, 2.5, 5, and 10 μg/mL) for 24 h. $n = 3$ independent samples. d Western blots show that the protein expression of STAT3, p-STAT3, PD-L1, and GAPDH in neutrophils is upregulated at different time points (0.5, 6, and 12 h) after SAA exposure (1 μg/mL). $n = 3$ independent samples. e The six-plex mIHC assay shows the co-expression of SAA (red), p-STAT3 (orange), and PD-L1 (green) in CD15⁺ (yellow) neutrophils from HCC peritumoral specimens. $n = 18$ patients. f In neutrophils exposed to SAA (1 μg/mL) for 24 h, napabucasin/BBI608, a small molecular STAT3 inhibitor, can significantly reduce the PD-L1⁺ neutrophils via a dose-dependent manner (0.01, 0.1, and 1 μmol/L). $n = 6$ biologically independent samples. Statistical data presented in this figure show mean ± SEM. ns indicates $P > 0.05$, *$P < 0.05$, **$P < 0.01$ and ****$P < 0.0001$, by one-way ANOVA (a, b, f), or two-sided Student's $t$ test (e). Source data and exact $P$ values are provided as a Source Data file.

cells for seven days were intraperitoneally injected with anti-SAA1/2 antibody, anti-PD-1 antibody, or the combined treatment every four days (Fig. 6a). A synergistic effect was observed in the combination of anti-SAA and anti-PD-1 group (Fig. 6b, c). The liver tumor weights and the ratio of liver to mouse body weights were significantly decreased after being treated with the combined treatment. The mice's body weight in the combined treatment group was stable and comparable to those in the control and single treatment groups (Fig. 6d, e). Dual-blockade of SAA and PD-1 not only reduced the PD-L1⁺ neutrophils (PD-L1⁺CD11b⁺Ly6G⁺ cells) infiltration (Fig. 6f, Supplementary Fig. 13 and 14a, b) but also enhanced the infiltration of TNF⁺/IFN-γ⁺CD8⁺ T cells (Fig. 6f and Supplementary Fig. 14c–f) in peritumoral tissue of HCC by using the flow cytometry analysis. The confocal microscopy images showed that PD-L1 expression on neutrophils could be inhibited by SAA1/2 blockade in the peritumoral region in vivo (Fig. 6g). To further validate the observed effect, we performed similar experiments using a spontaneous HCC model with *c-myc* overexpression and *p53* knockout background via hydrodynamic injection of corresponding plasmids. Importantly, the synergistic effect was also observed in the spontaneous HCC mouse model (Supplementary Fig. 15).

Besides, anti-PD-1 immunotherapy significantly shrunk the primary tumor and reduced lung metastasis in *Saa1* ablation (*Saa1*⁻/⁻) mice that were orthotopically implanted with hepa1-6-luci⁺ cells in the liver (Fig. 7a–c and Supplementary Fig. 14g). Furthermore, in confocal microscopy images of double-stained sections of liver samples with anti-mouse MPO antibody, anti-CD8 antibody and anti-mouse IFNγ antibody, *Saa1*⁺/⁺ images showed enrichment of MPO⁺ neutrophils in the peritumoral region but lack of IFNγ⁺ CD8⁺ T cells, while *Saa1*⁻/⁻ images showed the opposite results (Fig. 7d–f).

### STAT3 deficiency or inhibition enhances efficacy of anti-PD-1 based therapy in vivo

To further validate the significance of the STAT3 pathway in hepatocytes in mediating anti-PD-1 resistance in HCC, we used *Stat3*^flox/flox^*Alb-cre*⁺ mice (experiment) which lacked *Stat3* in hepatocytes and *Stat3*^flox/flox^*Alb-cre*⁻ mice (control) to build HCC models. After three doses of the PD-1 neutralizing antibody, the liver tumor was significantly inhibited in the *Stat3*^flox/flox^*Alb-cre*⁺ HCC model as compared to the *Stat3*^flox/flox^*Alb-cre*⁻ model (Fig. 7g). There was no difference in the body weight of the mice between the experiment and control group. However, both the liver tumor weight and the ratio of liver to mice body weight were significantly reduced in the experiment group, indicating the synergistic anti-HCC activity of the combination of STAT3 pathway deficiency and PD-1 checkpoint-blockade (Fig. 7h). In the peritumoral region of *Stat3*^flox/flox^*Alb-cre*⁺ mice, the infiltration of PD-L1⁺MPO⁺ neutrophils were less than in the control model (Fig. 7i, k). Besides, the confocal microscopy images showed enriched IFNγ⁺CD8⁺ T cells but a lack of MPO⁺ neutrophils in the peritumoral region of *Stat3*^flox/flox^*Alb-cre*⁺ mice, while the opposite distribution of the above cells was observed in *Stat3*^flox/flox^*Alb-cre*⁻ mice (Fig. 7j, k).

As to explore whether STAT3 inhibition has a synthetic effect with the anti-PD-1 agent in the liver orthotopic HCC mice model, we used napabucasin (BBI608), a small molecule inhibitor of STAT3 phosphorylation. The results showed that the combination of STAT3 inhibition and anti-PD-1 agents was safe (without affecting the mice body weight) and effective in vivo (Fig. 8a–e). We subsequently harvested Hepa1-6 samples and found that BBI608 could reduce the synthesis of SAA protein and recruit neutrophils (Ly6G⁺ cells) in the peritumoral area using multiplexed quantitative and spatially resolved immunofluorescence analyses (Fig. 8f). Flow cytometry analysis showed a significantly decreased infiltration of (CD11b⁺Ly6G⁺ cells) neutrophils and PD-L1⁺ neutrophils (PD-L1⁺CD11b⁺Ly6G⁺ cells) (Fig. 8g and Supplementary Fig. 16a, b), and PD-1⁺CD8⁺ T cells in the group of dual blockades of STAT3 and PD-L1/PD-1 pathway (Fig. 8g and Supplementary Fig. 16c–e).

In this study, we also observed that a triple-combination of STAT3, PD-1, and tyrosine kinase deficiency was more effective in vivo than a dual-combination of PD-1 and tyrosine kinase impairment (Supplementary Fig. 16f, g). However, the weight of the mice was not altered, indicating the safety of triple-combination (Supplementary Fig. 16h).

## Discussion

In order to stratify patients and enhance immunotherapy-based treatments, it is crucial to discover biomarkers and comprehend how they govern resistance to immune checkpoint inhibition. Here, we show that hepatocyte-derived SAA increases the expression of PD-L1 in neutrophils, promoting anti-PD-1 immunotherapy resistance in HCC. In addition, high levels of SAA in circulation are associated with poor clinical outcomes in aHCC patients treated with a single anti-PD-1 drug or combination therapy.

Currently, biomarkers for predicting PD-1 antibody response refer to the landscape of the immune cells in the microenvironment, molecular features of cancer cells, and systemic inflammation factors[10,28–31]. Local markers are generally regarded to be more accurate and powerful predictors because they directly reflect the immune status of TME. However, there is a lack of authoritative predictors for immunotherapy in HCC, and the predictive value of many classic indicators in different cohorts is controversial. For example, objective responses occurred in anti-PD-1/PD-L1 immunotherapies regardless of PD-L1 expression on tumor cells according to the post-hoc biomarker analysis in large-scale clinical trials[4,32]. Inevitably, tissue biopsy cannot reflect the entire microenvironment status of aHCC due to the spatially intratumoral or intertumoral heterogeneity of extensive diseases[33]. Besides, the varying predictive sensitivity of PD-L1 expression may exist in different PD-L1⁺ cell subtypes influenced by the diverse microenvironment composition[34,35]. Another confounding factor is that there is no widely accepted technical consensus for the immunohistochemistry analysis of PD-L1 expression in terms of tissue processing, antibody choosing, and the scoring of immune cell PD-L1 expression by pathologists[36,37].

Peripheral biomarkers, such as serum AFP or inflammatory cytokines, are suggested as more convenient and predictive indicators for systematic treatment in aHCC[10,38]. In our study, through the testing of

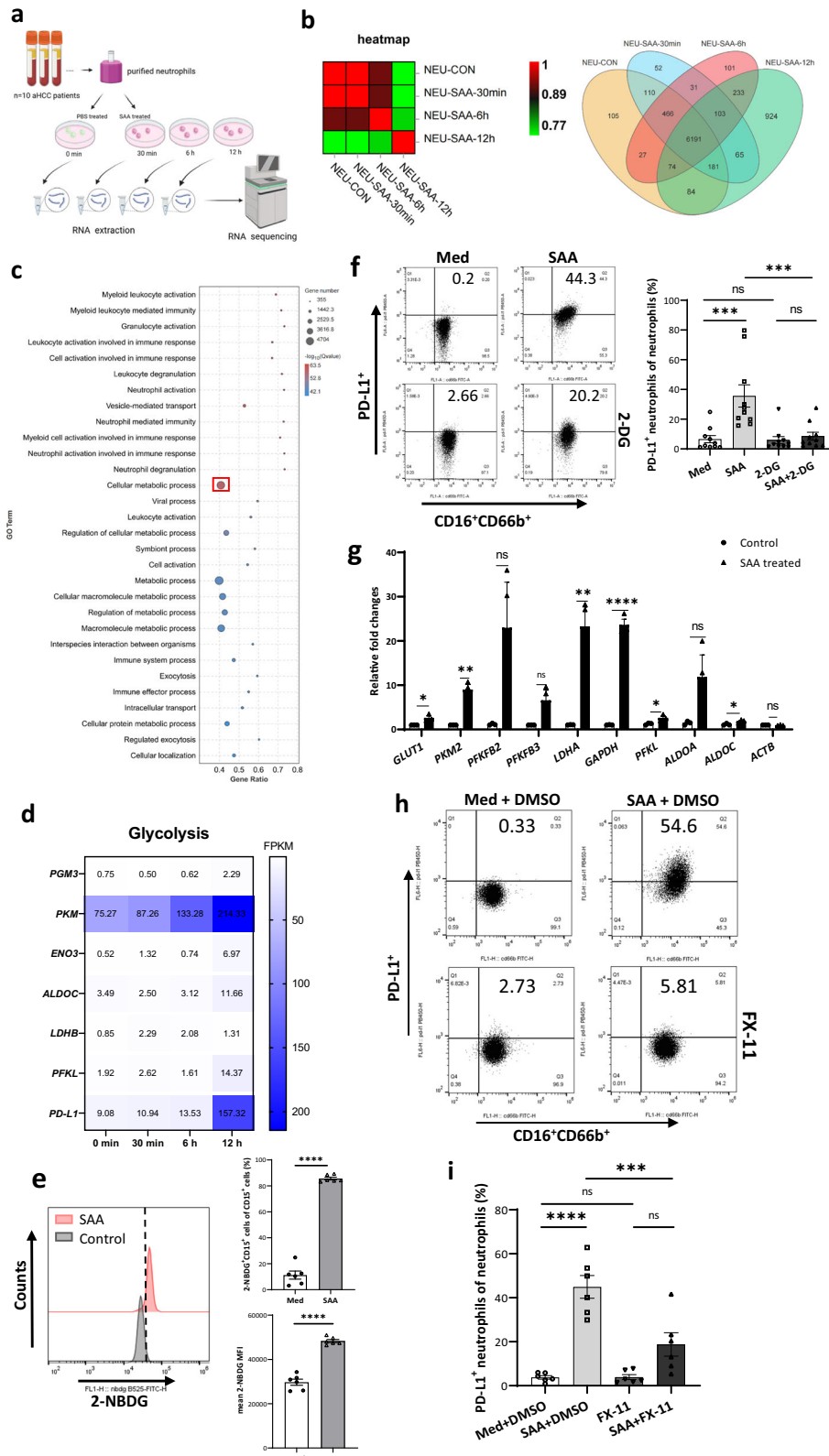

16 biomarkers (i.e., AFP, PIVKA-II, and inflammatory cytokines) in peripheral blood of aHCC patients treated with anti-PD-1 immunotherapy, we found that inflammatory-associated factor-SAA correlates with dismal response and survival. As a peripheral inflammatory biomarker, circulating SAA is highly correlated with peritumoral SAA, thus conveniently and consistently reflecting the local tumor status to predict the effect of immunotherapy. AFP, GPC3, CRP, and other peripheral blood proteins have been reported as biomarkers for predicting HCC survival or treatment effect, but in our study cohort, we found that baseline circulating SAA level had a higher AUC value for predicting PD status than these peripheral factors, as well as NLR and PLR[10,39]. It should be noted that our study is conducted in a retrospective cohort, which needs further validation in prospective studies with larger sample sizes.

**Fig. 4 | LDHA mediates glycolysis-induced PD-L1 expression on neutrophils.**
**a** Neutrophils purified from the peripheral blood of aHCC patients are treated with SAA (1 μg/mL) for 30 min, 6 h, and 12 h and subsequently used for RNA-seq. $n = 10$ biologically independent samples. **b** The heatmap and Venn diagram of transcriptomic data is generated by analyzing the gene expression of each group. **c** GO term analysis suggests that DEGs are enriched in terms of cellular metabolic processes. **d** The levels of glycolysis-related gene expression were assessed by the transcriptomic data of SAA treated-neutrophil. **e** 2-NBD-glucose consumption of neutrophils is significantly increased after co-culture with SAA (1 μg/mL) for 12 h in vitro by the flow cytometry analysis. $n = 6$ biologically independent samples. **f** The glycolysis inhibitor-2DG (25 mmol/L) was used to treat neutrophils for 12 h. Dot plots and statistical analysis show that PD-L1 expression on SAA (1 μg/mL)-induced cells is significantly reduced by 2DG. $n = 10$ biologically independent samples.

**g** Neutrophils purified from peripheral blood aHCC patients ($n = 10$) are treated with SAA (1 μg/mL) for 0.5, 6, and 12 h. The levels of glycolysis-related gene expression were quantified by qPCR. The mRNA expression of glycolytic enzymes including *GLUT1, LDHA, GAPDH, PFKL, ALDOA*, and *ALDOC* is increased in SAA-treated neutrophils by RNA-seq analysis. **h, i** The LDHA inhibitor-FX-11 (50 mmol/L) was used to treat neutrophils for 12 h. The apoptosis of neutrophils is analyzed by flow cytometry which reveals that PD-L1 expression on SAA (1 μg/mL)-induced cells is significantly reduced by FX-11. $n = 6$ biologically independent samples. Statistical data presented in this figure show mean ± SEM. ns indicates $P > 0.05$, *$P < 0.05$, **$P < 0.01$, ***$P < 0.001$, and ****$P < 0.0001$, by two-sided Student's $t$ test (**e**, **g**), or one-way ANOVA (**f**, **i**). Source data and exact $P$ values are provided as a Source Data file. Illustrations created with BioRender.com.

An in-depth study of the mechanisms of SAA regulation helps provide targeted treatment for these patients. Our study revealed that activation of STAT3 induced the secretion of SAA to the periphery of the tumor by hepatocytes, which was consistent with previous reports[15]. The regulation of peritumoral hepatocytes on the tumor microenvironment is an important part of the organ specificity of liver tumors[40]. Furthermore, a previous study reported that liver metastases could diminish immunotherapy efficacy and act as the independent biomarker of PD-1 response, which highlights the role of specific organic microenvironment in PD-1 resistance[41]. Although there are differences in the peritumoral hepatocytes microenvironment in patients with different etiologies such as hepatitis virus infection, alcohol abuse, and metabolic disorders, investigating the interactions between tumor cells and hepatocytes is helpful for a deeper understanding of the liver-specific tumor microenvironment[42,43]. Through the detection of tumor immune landscape, we found that SAA mainly recruits neutrophils, induces its PD-L1 expression by regulating the glycolytic metabolism, further inhibits T cells' cytotoxicity from exerting an immunosuppressive effect, and promotes immune escape in aHCC. Previous studies have reported that PD-L1$^+$ macrophages contribute to anti-PD-1 resistance[44]. However, the expression level of SAA was remarkably associated with PD-L1$^+$CD15$^+$ neutrophils but not PD-L1$^+$CD68$^+$ macrophages, according to the six-plex mIHC analysis in our study, indicating that the mechanisms of PD-1 antibody resistance are diverse, and various immune cells are involved in mediating immune response in tumors with different biomarkers expression. Therefore, our study may supplement the understanding of the regulatory mechanisms of the immune microenvironment in the liver and the response to immunotherapy.

SAA regulates the immunosuppressive function of neutrophils via glycolysis, in which several enzymes are potential targets. Our study found that LDHA played a key role in SAA-induced PD-L1 expression on neutrophils, so targeting LDHA may promote antitumor immunity in patients with high baseline circulating SAA expression. Besides, the expression of key glycolysis-related enzymes in HCC is highly correlated with STAT3[45,46]. We thus detected the roles of STAT3 in the process of SAA-induced PD-L1 and found that the process also depended on STAT3 activation. Therefore, targeting STAT3 by BBI608 can inhibit the secretion of SAA derived from hepatocytes and reduce PD-L1 expression on SAA-stimulated neutrophils. In vivo, we also demonstrated the antitumor effect of BBI608 in combination with PD-1 antibodies, as well as its role in the reduction of tumor-associated neutrophil infiltration and PD-L1 expression. Notably, we also examined the antitumor effects of SAA blockade by BBI608 combined with lenvatinib and PD-1 antibody and found that the triple therapy further improved the efficacy of the combination therapy of tyrosine kinase inhibitor with immunotherapy. Therefore, SAA can potentially be a valuable biomarker in the era of anti-PD-1-based combinations for aHCC.

In conclusion, we investigated a panel of serum biomarkers related to anti-PD1 response and identified inflammatory factor-SAA as a significant biomarker and prospective immunotherapy target in aHCC. Our findings show that hepatocyte-derived SAA attracted and increased PD-L1 expression on neutrophils via LDHA/STAT3, impairing T cell cytotoxicity and mediating PD-1 antibody resistance (Fig. 8h). We primarily prove a liver-specific mechanism, classified individuals who will benefit from PD-1 blocking and propose new strategies for HCC patients undergoing anti-PD-1 immunotherapies to improve clinical outcomes.

## Methods
### Ethical issues
The study was approved by the Ethical Review Committee of Sun Yat-sen University Cancer Center (SYSUCC, Guangzhou, China; approval number: GZR2017-236 and B2023-688-01). All the patients who took part in the trial provided written informed consent. All mice were housed and treated in the animal facility of Sun Yat-sen University Cancer Center. All animal procedures have been done according to the institutional guidelines and approved by the Sun Yat-sen University Cancer Center ethics committee (approval number: L102012017003I).

### Patients' blood and tissue samples
Sequential blood samples were obtained from 326 patients with aHCC who underwent anti-PD-1 immunotherapy alone or anti-PD-1 in combination with tyrosine kinase inhibitors (TKIs) locoregionally between April 2017 and December 2019. Fifty-two samples from patients treated with anti-PD-1 monotherapy were analyzed for 16 biomarkers, while the remaining 274 samples from patients treated with combined therapies were examined for SAA levels (Supplementary Fig. 1a and Supplementary Table 1, 4). In addition, between February 2020 and July 2022, fresh blood samples from 68 patients with aHCC were collected for in vitro co-culture systems, flow cytometry, RNA-sequencing (RNA-seq), and western blotting assays (Supplementary Table 7). Further, formalin-fixed, paraffin-embedded samples of HCC patients receiving anti-PD-1 monotherapy and tissue microarray (TMA) of resected HCC tumors without anti-PD-1 treatment were utilized for multiplexed immunohistochemistry (mIHC) analysis. The TMA, comprised of tumor cells and peritumoral tissues containing hepatocytes, was constructed from 160 resected HCC samples collected at the SYSUCC Tumor Biobank between November 2005 and November 2011 (Supplementary Table 6). Additionally, between October 2019 and July 2021, fresh core-needle biopsy tissues from 6 patients with aHCC were snap-frozen for RNA-seq, and all the lesions were firmly classified as HCC based on pathological examination (Supplementary Table 3).

### Patients and clinical follow-up
We retrospectively screened 569 patients who received anti-PD-1-based immunotherapies either as single or combined with tyrosine kinase inhibitors (TKIs), locoregional procedures, or both between

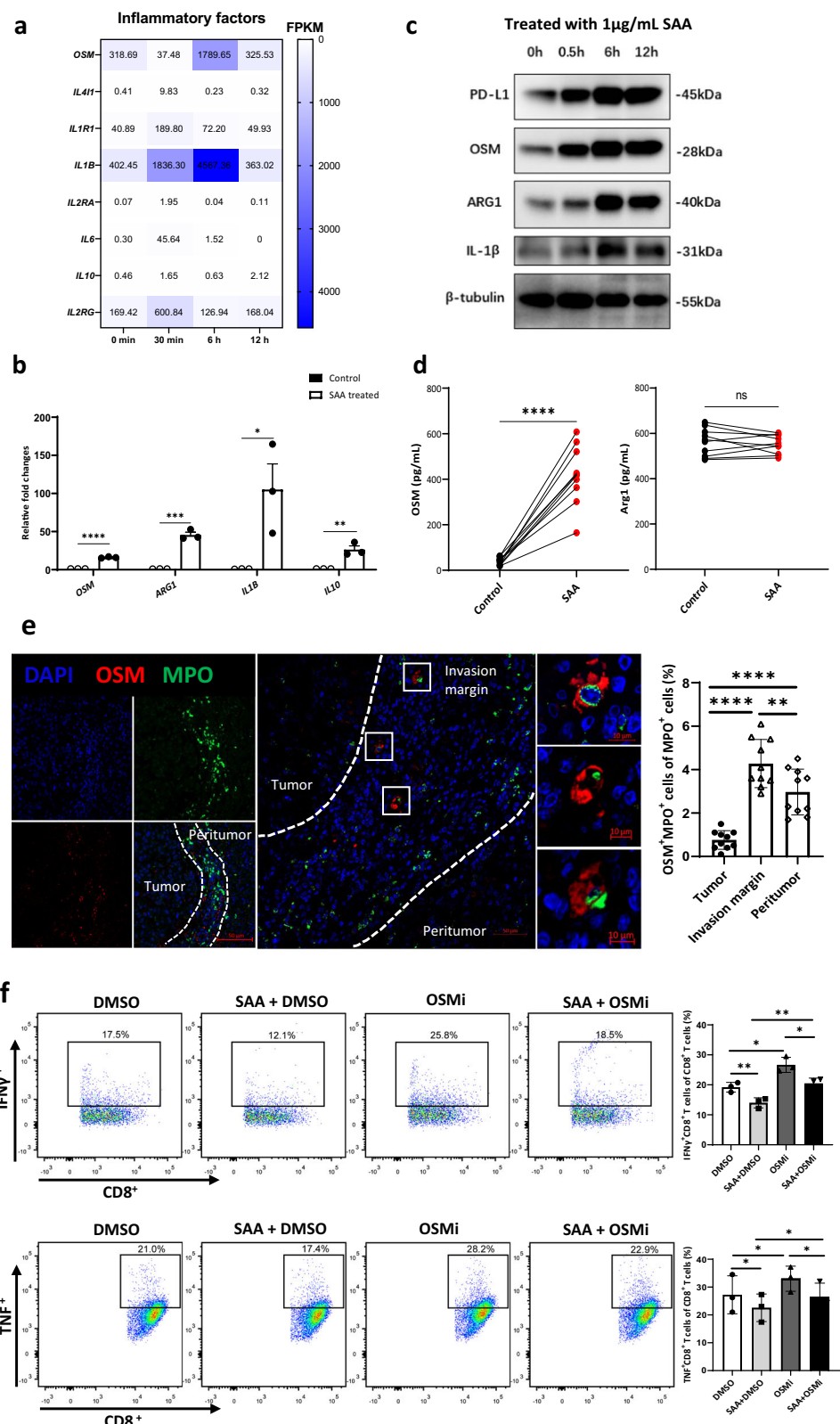

April 2017 and December 2019. A total of 243 patients were excluded because of missing image examination ($n = 109$), uncompleted follow-up (n = 79), without target tumors for radiological assessment ($n = 24$), missing written informed consents of patients ($n = 17$), secondary malignancies ($n = 9$), and treated with traditional Chinese medicine along with anti-PD-1 agent ($n = 5$). Among the remaining 326 patients, all of whom were obtained written informed consents, 52 patients were

treated with single PD-1 inhibitors, 99 patients were treated with PD-1 inhibitors plus TKIs, 97 patients were treated with PD-1 inhibitors plus locoregional procedures (including transarterial chemoembolization and transarterial infusion of chemotherapy), and 78 patients were treated with a combination of PD-1 inhibitors, TKIs, and locoregional procedures (Supplementary Fig. 1a). Furthermore, in order to validate our findings, we conducted a retrospective collection of data from 138

**Fig. 5 | SAA upregulates OSM production in neutrophils and reduces IFN-γ and TNF release in autologous CD8⁺ T cells. a, b** The mRNA expression of genes associated with N2 phenotype neutrophils is increased in SAA (1 μg/mL)-treated neutrophils including *OSM*, *ARG1*, *IL-1B*, and *IL-10* by qPCR. $n = 10$ biologically independent samples. **c** The correlated protein expression of genes including PD-L1, OSM, ARG1, and IL-1β is validated in SAA (1 μg/mL)-treated neutrophils by western blotting analysis. $n = 3$ independent samples (**d**) The level of OSM and ARG1 is determined in SAA (1 μg/mL)-treated neutrophils by ELISA. $n = 10$ biologically independent samples. **e** Paraffin-embedded HCC samples are double-stained with anti-human MPO antibody (green) and anti-human OSM antibody (red). $n = 10$

patients. The OSM⁺MPO⁺ cells are observed in the peritumoral region by confocal microscopy. **f** Neutrophils treated with medium, SAA (1 μg/mL), OSM-SMI-10B (50 μmol/L) and SAA (1 μg/mL) plus OSM-SMI-10B (50 μmol/L) for 12 h are used to co-culture with autologous CD8⁺ T cells. The levels of IFN-γ and TNF production in CD8⁺ T cells were analyzed after 12 h of co-culture by flow cytometry. $n = 3$ biologically independent samples. Statistical data presented in this figure show mean ± SEM. ns indicates $P > 0.05$, *$P < 0.05$, **$P < 0.01$, ***$P < 0.001$, and ****$P < 0.0001$, by two-sided Student's $t$ test (**b**, **d**, **f**), or one-way ANOVA (**e**). Source data and exact $P$ values are provided as a Source Data file.

patients who received anti-PD-1-based immunotherapies combined with tyrosine kinase inhibitors (TKIs), as well as locoregional procedures, or both, between January 2021 to January 2023 (Supplementary Table 5). Patient's epidemiologic characteristics, treatment strategies, and follow-up information were obtained from a prospective clinical database of SYSUCC. All therapies were performed according to the recommendations of clinical trials with high-level of evidence. Response assessment by computed tomographic or magnetic resonance imaging was performed every 6–8 weeks in the first year of treatment and every 8–12 weeks thereafter. After two years of anti-PD-1 treatment, patients were recommended to stop immunotherapy and contacted every 8–12 weeks to document the survival status. Survival outcomes were defined as follows: overall survival (OS) was defined as the time between randomization date and death of any cause and progression-free survival PFS was defined as the time between randomization date and disease progression or death of any cause. Radiological tumor response assessment was conducted according to the Response Evaluation Criteria in Solid Tumors (RECIST) version 1.1. The follow-up data of the derivation cohort was censored at the time of July 15, 2021, while the validation cohort was censored at the time of July 20, 2023.

### LASSO multivariate analysis
LASSO multivariate analysis was used to verify the predicting factors for tumor response (PD/NPD) among the 16 variables. The binomial deviance curve was plotted versus log $(\lambda)$, where $\lambda$ is a tuning hyperparameter. Solid vertical lines represent binomial deviance ± standard error (SEM). The dotted vertical lines are drawn at optimal values by using the minimum criteria and 1-SE criteria. An optimal $\lambda$ value was selected for the LASSO model by using 10-fold cross-validation via minimum criteria. A coefficient profile plot was produced against the log $(\lambda)$ sequence.

### Cell lines
Liver cell lines HHL-5 (kindly provided by Prof. Bo Li, Sun Yat-sen University) derived from human hepatocytes were used for in vitro experiments. The murine HCC cell lines Hepa1-6-luc⁺ were purchased form Procell used for in vivo experiments. Neutrophils or T cells were isolated from the blood of patients with advanced HCC (aHCC) treated at the Sun Yat-sen Cancer Center or healthy donors.

HHL-5 and Hepa1-6 cell lines were cultured in Dulbecco's Modified Eagle's Medium (Gibco, China). Neutrophils and T cells were cultured in RPMI-1640 (Gibco, China). Both media were supplemented with 10% fetal bovine serum (Hyclone, New Zealand) and 1% penicillin-streptomycin (Gibco, China). All cells were cultured at 37 °C and 5% $CO_2$.

### Blood samples collection and quantitative assays
Blood samples were collected routinely following the manufacturer's instructions and were tested within 24 h of sampling. Briefly, blood was centrifuged for 10 min at 800 x $g$ and then the serum was collected for further analysis. Plasma concentration of

SAA protein was measured using a commercially available Human serum amyloid A1/SAA1 ELISA kit (Biodragon, China; Cat# BDEL-0724-48T). Level of serum CRP and ALB were quantified using the Human CRP ELISA kit (KeyGen, USA; Cat# KGC1317-96) and the Human Albumin ELISA kit (Neobioscience, USA; Cat# EHC024.48), respectively. Blood routine examinations were performed with the SYSMEX XN-2000 automatic blood analyzer. All procedures are performed according to the manufacturer's instruction.

### Isolation and culture of neutrophils and CD8⁺ T cells
Neutrophils and CD8⁺ T cells were isolated from the peripheral blood of patients with aHCC or healthy donors by magnetic beads according to the manufacturer's instruction for use in subsequent in vitro experiments (for neutrophils: EasySep Direct Human Neutrophil Isolation Kit, STEMCELL Technologies, USA, Cat# 19666; for CD8⁺ T cells: EasySep Human CD8 Positive Selection Kit II, STEMCELL Technologies, USA, Cat# 17853).

Purified neutrophils were used for human APO-SAA protein, 2-DG, FX-11, OSM-SMI-10B, TEPP-46, and αKG stimulations and coculture analysis. For coculture of neutrophils and T cells, a total of $1 \times 10^6$ purified neutrophils were treated with control medium, SAA (1 μg/mL), OSM-SMI-10B (50 μmol/L), or SAA (1 μg/mL) plus OSM-SMI-10B (50 μmol/L) for 12 h, and then cocultured with autologous T cells at a 1:2 ratio for 12 h in a complete RPMI-1640 medium supplemented with 2 μg/mL anti-CD3 and 2 μg/mL anti-CD28 antibodies (Biolegend, USA; Cat# 317326 & 302934). The T cells were then collected for the expression of IFNγ and TNF analysis.

### Examination of the supernatant level of OSM and ARG1
Purified neutrophils were stimulated with DMSO, SAA (1 μg/mL), OSM-SMI-10B (50 μmol/L) or SAA (1 μg/mL) plus OSM-SMI-10B (50 μmol/L), and the supernatant level of OSM or ARG1 were examined by ELISA kits (RayBiotech, ELH-OSM-1, Cat# ELH-OSM-1 and ELH-ARG1-1, Cat# ELH-ARG1-1, respectively).

### Proteome profiler human inflammatory cytokine array
The serum isolated from the peripheral blood of patients with aHCC was used to determine the relative levels of TH1/TH2 cytokines by the Human TH1/TH2 Array 1 (RayBiotech, China; Cat# QAH-TH-1). Approximately 50 μL peripheral plasma collected was tested according to the manufacturer's protocol. The levels of the cytokines in this panel were analyzed by ScanArray Express (Perkinelmer, USA).

### Trypan blue exclusion assay
The trypan blue exclusion assay was used to determine cell viability. Neutrophils were centrifuged for 5 min at $100 \times g$, and then resuspended in 1 mL PBS buffer. The cells were then stained with 0.4% trypan blue (Thermofisher, USA) at approximately 27 °C for 5 min and unstained cells were counted using a Cell Counter (OLYMPUS, Japan). The data was expressed as the percentage of unstained cells compared with the control cells not exposed to SAA (1 μg/mL) or other chemical reagents.

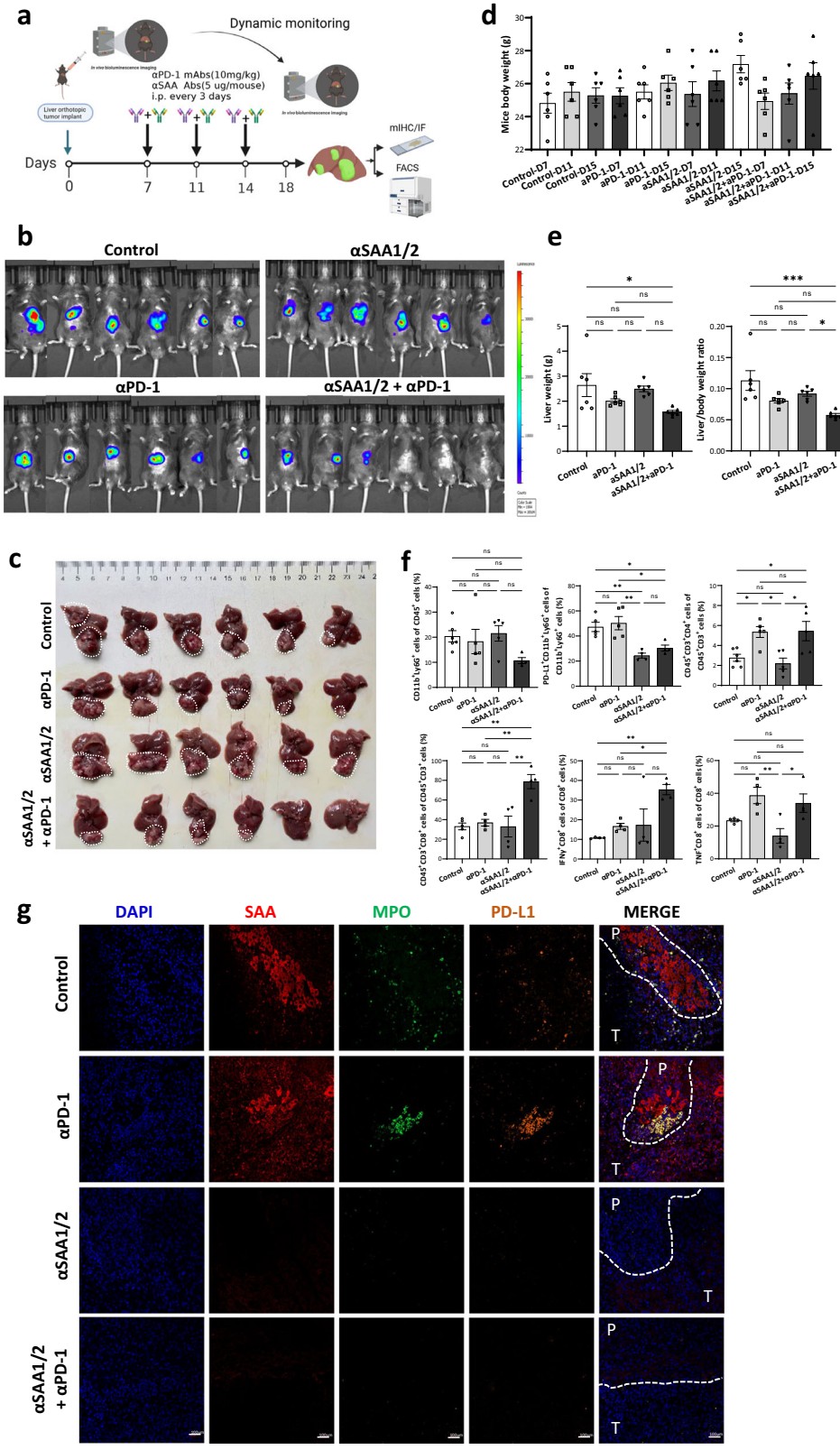

## Neutrophil apoptosis assay

A total of $1 \times 10^6$ neutrophils were cultured in complete RPMI-1640 media supplemented with or without SAA (1 μg/mL) protein for 24 h and then were harvested. Neutrophil apoptosis assay was performed using Annexin V Apoptosis Detection Kit I (BD Biosciences, USA; Cat# 556547) according to the manufacturer's instructions.

## Quantitative real-time PCR

Total RNA was extracted using RNA Quick Purification kit (ESscience, China; Cat# RN001) and then 1 μg total RNA was used to synthesize cDNA with 5×All-In-One RT MasterMix (Vazyme, China; Cat# R233-01) according to the protocol. Diluted cDNA was subjected to quantitative real-time PCR according to a standard protocol using the SYBR Green

**Fig. 6 | αSAA1/2 enhances αPD-1 efficacy in vivo. a** Orthotopic liver murine models implanted with Hepa1-6-luci$^+$ cells were separately treated with PBS combined with 0.5% methylcellulose, αPD-1 (10 mg/kg), αSAA1/2 (5 μg/mice), or αPD-1 (10 mg/kg) combined with αSAA1/2 (5 μg/mice) per three days. $n = 6$ mice. After 18 days, the mice were sacrificed and necropsied. The whole body and organs of mice were weighed after treatment. The dissected tumor tissues were prepared for histological examination, FACS, and immunofluorescence analysis. **b** Before sacrifice, luciferase marker expression was detected using IVIS. **c** Images of tumor tissue samples were taken from the necropsied orthotopic mice model. **d, e** Histograms showed weight changes in the whole mouse, liver tumor, and liver to mouse ratio in different treatment groups. $n = 6$ mice. **f** Tumor infiltration of neutrophils (CD11b$^+$Ly6G$^+$ cells), PD-L1$^+$ neutrophils (PD-L1$^+$CD11b$^+$Ly6G$^+$ cells), CD45$^+$CD3$^+$CD4$^+$ T cells, CD45$^+$CD3$^+$CD8$^+$ T cells, IFNγ$^+$CD8$^+$ T cells, and TNF$^+$CD8$^+$ T cells were analyzed by FACS in each group. $n = 6$ mice. **g** The confocal microscopy images showed the co-expression of SAA, PD-L1, MPO$^+$ neutrophils in each treatment group by immunofluorescence analysis. DAPI: blue, MPO: green, PD-L1: orange, and SAA: red. Statistical data presented in this figure show mean ± SEM. ns indicates $P > 0.05$, *$P < 0.05$, **$P < 0.01$, and ***$P < 0.001$, by one-way ANOVA (**e, f**). Source data and exact $P$ values are provided as a Source Data file. Illustrations created with BioRender.com.

Real-Time PCR MasterMix (Vazyme, China; Cat# Q311-02) in a Roche LightCycler 480 System (Basel, Switzerland). Sequences of the primers used for the PCR analysis are listed in Supplementary Table 9. To determine the relative fold changes of different genes, their levels of expression were normalized to those of ACTB or 18 S rRNA.

## Western blot assay
Cells were washed with ice-cold PBS and then lysed in lysis buffer with added 1 mmol/L phenylmethyl sulfonyl fluoride (PMSF) (Beyotime, China; Cat# ST506), protease inhibitor cocktail (Cwbio, China; Cat# CW2200S), and phosphatase inhibitor cocktail (Cwbio, China; Cat# CW2383S) for 5 min. Following this, the lysis was boiled for 10 min at 98–100 °C. Protein concentrations were determined by BCA Protein Assay Kit (Thermo Fisher Scientific, USA; Cat# 71285-3). The primary antibodies used were summarized in Supplementary table 8. Images were prepared using ChemiDoc Touch (BioRad, China) and analyzed with Image Lab software (BioRad, China).

## Metabolic quantification
The rate of glucose uptake and lactate secretion was quantified using Multiparameter Bioanalytical System (YSI 2900, YSI Life Sciences) in 96-well format and analyzed.

## Seahorse
Glycolysis of neutrophils was analyzed through the Seahorse XFe96Analyzer (Agilent, California, USA). The XF96-well cell culture plate (Agilent, USA) was coated with 1 mg/mL polylysine and incubated at 37 °C overnight. Next day, $1 \times 10^4$ fresh neutrophils purified from 5 aHCC patients were treated with control medium or SAA (1 μg/mL) and then seeded in a medium of bio-carbonate free DMEM with 1 mmol/L glutamine. They were then incubated at 37 °C for 60 min in the $CO_2$ free incubator to balance the media pH and temperature. Consecutive measurements were obtained under basal conditions and after the sequential addition of 10 mmol/L glucose (inhibiting mitochondrial ATP synthase), 2 μmol/L oligomycin (revealing cellular maximum glycolytic capacity), and 50 mmol/L 2-deoxy-glucose (2-DG, A glucose analog that inhibits glycolysis by competitively binding glucohexokinase).

## RNA isolation, sequencing, and data analysis
Tumor samples for RNA sequencing (RNA-seq) were immediately snap-frozen in liquid nitrogen after harvesting. Neutrophils were purified from peripheral blood of 10 aHCC patients and treated with 1 μg/mL SAA protein for 30 min, 6 h and 12 h, and then harvested for RNA-seq. Total RNA was extracted as described above, immediately after harvesting. RNA-seq was performed by Genedenovo (Genedenovo, China). Raw mapped reads were processed in R to determine differentially expressed genes and generate normalized read counts. Differential expression analysis was performed using the DESeq2 R package (1.16.1). Principal Component Analysis (PCA) was performed by the ggplot2 R package. Gene ontology enrichment analysis was implemented by the cluster Profiler R package. Gene ontology terms with corrected $P < 0.05$ were considered significantly enriched by differentially expressed genes.

## Multi-colored immunohistochemistry
Formalin-fixed paraffin-embedded human and mouse liver sections were collected. Multiplex immunofluorescence staining was performed on the PANO 7-plex IHC kit (Panovue, China; Cat# 10004100050) according to the manufacturers' instructions. Primary antibodies dilution and incubation were carried out simultaneously using the optimal dilution for each assay and further details regarding the antibodies used in the assay, please refer to Supplementary Table 8. Tissue sections were incubated with fluorophores Opal 520, 570, 690, 620, 480, and 780 for 10 min at room temperature. Antigen retrieval was performed using 10 mmol/L sodium pH 6 citrate buffer or EDTA pH 8.0/9.0 in a microwave for 5–10 min. Finally, all sections were counter-stained with DAPI. Images were captured using Vectra Polaris Automated Quantitative Pathology Imaging System (Akoya Biosciences, USA). Positive stain quantification and spatial cell analysis were performed using HALO image analysis software (Indica Labs, USA, version 2.3.2089.34).

## Multiplexed immunofluorescence analysis
Mice liver tumor tissue sections were deparaffinized, and antigens were recovered with EDTA (1 mmol/L, pH 8.0/9.0) and boiled for 20 min at 97 °C. Endogenous peroxidase was inactivated for 10 min, and the slides were incubated with a blocking solution containing 0.3% bovine serum albumin in 0.05% Tween-20 and Tris-buffered saline for 30 min. Primary antibody dilution and incubation were carried out simultaneously using the optimal dilution for each assay. The secondary antibodies and fluorescent reagents used were shown in Supplementary Table 8. The super-resolution images were obtained by Airyscan 2 with a confocal laser scanning microscope (LSM980, ZEISS; 63×/1.40 oil objective with 2.5x digital zoom, 35 nm/px, XY < 120 nm, Z < 350 nm).

## Tumor isolation and flow cytometry
For flow cytometry for cell lines, neutrophils and T cells were washed by pre-cold PBS and suspended in cell staining buffer (2% FBS in PBS). For flow cytometry for tumor tissues, tumors were harvested at 18 days after therapy. 1/3 of the tissues were digested into single-cell suspensions, and 1/3 of the tissues were prepared for histopathological assessment, while the left 1/3 of the tissues were immediately put in liquid nitrogen and then stored at −80 °C. The tumors were digested into single-cell suspensions. The cell suspensions were passed through a 70 μm cell strainer (Corning, USA), centrifuged at $300 \times g$ for 5 min at 4 °C and then suspended in 100 μL cell staining buffer.

Flow cytometry was carried out with the detailed procedure. Cells were blocked with CD16/32 antibody (Biolegend, USA; Cat# 302012). The cells were then incubated with the indicated antibodies for 30 min on ice. For intracellular staining of IFNγ and TNF, cells were stimulated in vitro with a cocktail containing phorbol 12-myristate 13-acetate (PMA), ionomycin, brefeldin A, and monensin (Biolegend, USA) for 4 h before surface staining. T Cells were then performed surface staining and incubated with 1 × Fixation/Permeabilization Solution (BD Pharmingen, USA) for 20 min at 4 °C

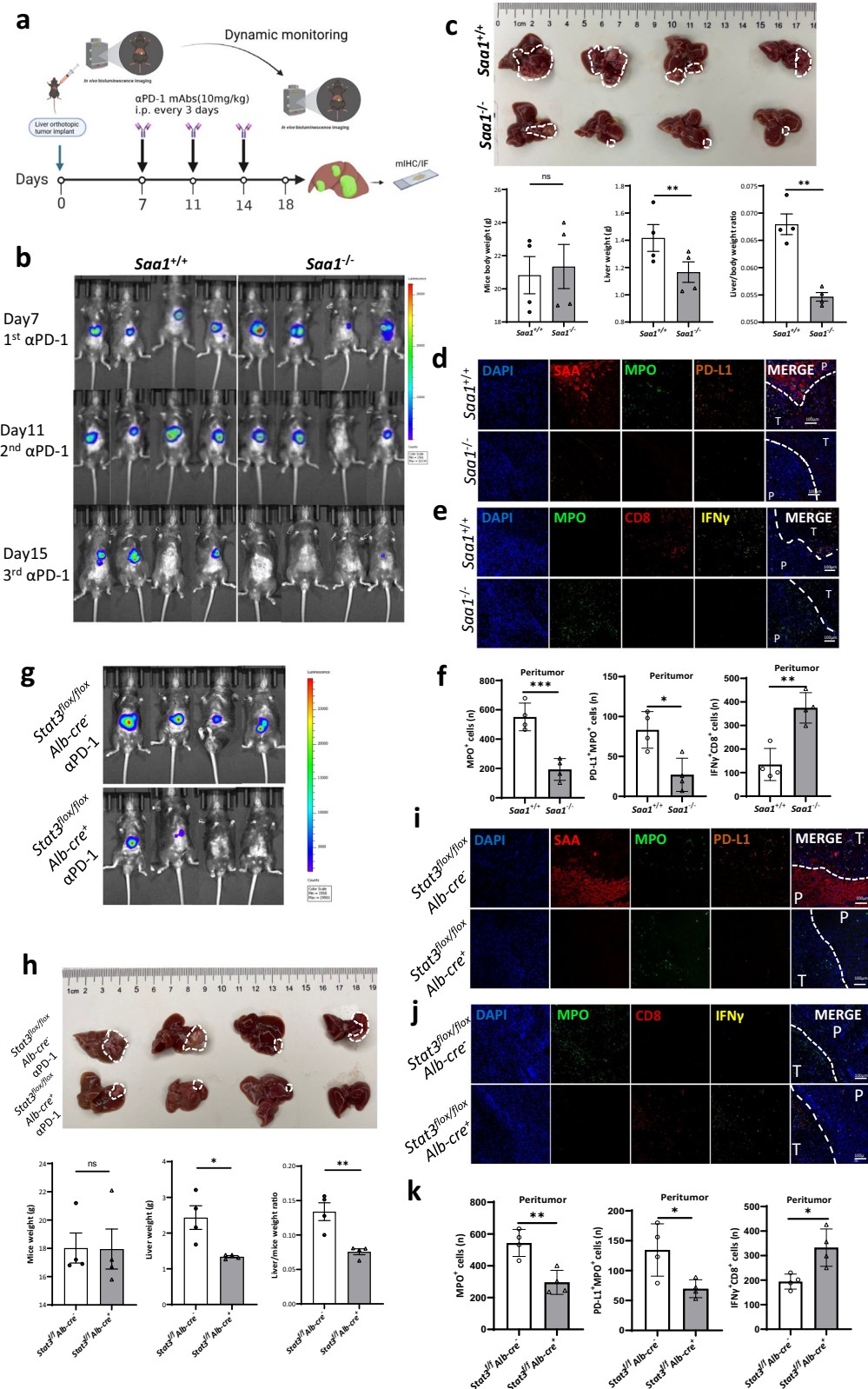

before intracellular staining. The fluorescent-labeled antibodies used are summarized in Supplementary Table 8. For 2-NBDG analysis, purified neutrophils were incubated with a labeled isoform of glucose 2-NBDG for 30 min at 37 °C before being subjected to flow cytometry analysis. Data were obtained with the Gallios analyzer (Beckman, USA) and evaluated with FlowJo software version V10 (TreeStar, USA).

**Mice**

WT male and female C57BL/6 J mice were purchased from the Guangdong Medical Laboratory Animal Center, and *Saa1* knockout mice were purchased from Cyagen Biosciences Inc (Guangzhou, China).

*Stat3*^flox/flox^ mice were purchased from Jackson Laboratory and bred to *Alb-cre*^+/+^ mice to generate *Stat3*^flox/flox^ & *Alb-cre*^+/−^ mice.

**Fig. 7 | αPD-1 shows active efficacy in *Saa1*<sup>-/-</sup> and *Stat3*<sup>flox/flox</sup>*Alb-cre*<sup>+</sup> (lacked *Stat3* in hepatocytes) orthotopic HCC mouse models. a** Hepa1-6-luci<sup>+</sup> cells were orthotopic implanted in the liver of *Saa1*<sup>+/+</sup>, *Saa1*<sup>-/-</sup> mice, *Stat3*<sup>flox/flox</sup>*Alb-cre*<sup>+</sup>, and *Stat3*<sup>flox/flox</sup>*Alb-cre*<sup>-</sup> mice, separately. *n* = 4 mice. Mice were treated with αPD-1 (10 mg/kg) for three days. Meanwhile, D-luciferin (100 mg/kg) was injected intraperitoneally for detecting bioluminescence in vivo by IVIS every four days. After 18 days, the mice were sacrificed. The whole body and liver of mice were weighed. The dissected tumor was prepared for histological examination and immuno-fluorescence analysis. **b** Luciferase was detected on day 7 (1st αPD-1), 11 (2nd αPD-1), and 15 (3rd αPD-1) of *Saa1*<sup>+/+</sup> and *Saa1*<sup>-/-</sup> mice. *n* = 4 mice. **c** Images of tumor were taken from *Saa1*<sup>+/+</sup> and *Saa1*<sup>-/-</sup> mice. Histograms showed weight changes in the whole mice, liver tumor, and liver to mouse ratio in different groups. *n* = 4 mice. The confocal microscopy images showed the co-expression of SAA, PD-L1, MPO<sup>+</sup> neutrophils (**d**), IFNγ<sup>+</sup>CD8<sup>+</sup> T cells and MPO<sup>+</sup> neutrophils (**e**) and quantification of MPO<sup>+</sup> cells, PD-L1<sup>+</sup>MPO<sup>+</sup> cells and IFNγ<sup>+</sup>CD8<sup>+</sup> T cells in peritumor (**f**) in *Saa1*<sup>+/+</sup> and *Saa1*<sup>-/-</sup> mice by immunofluorescence analysis. *n* = 4 mice. **g** Luciferase was detected on day 15 (3rd αPD-1) of *Stat3*<sup>flox/flox</sup>*Alb-cre*<sup>+</sup> and *Stat3*<sup>flox/flox</sup>*Alb-cre*<sup>-</sup> mice. **h** Images of tumor tissue were taken from *Stat3*<sup>flox/flox</sup>*Alb-cre*<sup>+</sup> and *Stat3*<sup>flox/flox</sup>*Alb-cre*<sup>-</sup> mice. Histograms showed weight changes in the whole mouse, liver tumor, and liver to mouse ratio in different groups. *n* = 4 mice. The confocal microscopy images showed the co-expression of SAA, PD-L1, MPO<sup>+</sup> neutrophils (**i**), IFNγ<sup>+</sup>CD8<sup>+</sup> T cells and MPO<sup>+</sup> neutrophils (**j**) and quantification of MPO<sup>+</sup> cells, PD-L1<sup>+</sup>MPO<sup>+</sup> cells and IFNγ<sup>+</sup>CD8<sup>+</sup> T cells in peritumor (**k**) in *Stat3*<sup>flox/flox</sup>*Alb-cre*<sup>+</sup> and *Stat3*<sup>flox/flox</sup>*Alb-cre*<sup>-</sup> mice by immunofluorescence analysis. *n* = 4 mice. CD8: dark red, DAPI: blue, IFNγ: yellow, MPO: green, PD-L1: orange, and SAA: brilliant red. The results are expressed as the mean ± SEM. Statistical data presented in this figure show mean ± SEM. ns indicates $P > 0.05$, *$P < 0.05$, **$P < 0.01$, and ***$P < 0.001$, by two-sided Student's *t* test (**c**, **f**, **h**, **k**). Source data and exact *P* values are provided as a Source Data file. Illustrations created with BioRender.com.

All transgenic mice were bred and maintained in the animal facility of the SYUCC or SYSU. All mice were maintained under specific pathogen-free (SPF) conditions and were used between 6 and 8 weeks of age. All mice were housed five per cage under a 12-h light-dark cycle (light on from 7 a.m. to 7 p.m., humidity between 30% and 70%, temperatures of 20–22 °C) with free access to food and water. Mice were monitored three times per week for general health and euthanized early based on defined endpoint criteria, including ascites, lethargy, or other signs of sickness or distress. If animals appeared moribund or the diameter of the tumors reached 15 mm, the mice were sacrificed. In some cases, the maximal tumor burden permitted has been exceeded the last day of measurement and the mice were immediately euthanized. Mice were age- and gender-matched with appropriate control mice for analysis.

## Orthotopic HCC model

For the orthotopic hepatic tumor model, a total of $1 \times 10^6$ luciferase-expressing murine HCC cells (Hepa1-6-luci<sup>+</sup>) were suspended in 25 μL of 66.66% Matrigel (Corning, China; Cat# 356234) and intra-hepatically injected into the left lobe of the liver of anesthetized 6–8-week-old C57BL/6 J mice. Seven days after orthotopic liver transplantation, the mice were randomized into three treatment groups and one control (*n* = 6–8 for each group): (1) intraperitoneal injections of vehicle (PBS with 5% DMSO); (2) aPD-1 antibody (1 mg/mL), 10 mg/kg two times per week; (3) anti-SAA1/2 antibody (25 μg/mL), 5 μg per mouse, two times/week; or napabucasin (BBI608), 10 mg/kg daily; Or lenvatinib, 10 mg/kg daily; and (4) anti-SAA1/2 antibody or napabucasin or lenvatinib and aPD-1 antibody, respectively. Lenvatinib (Cat# S1164, S5240) and napabucasin (STAT3 inhibitor, BBI608, Cat# S7977) were purchased from Selleck (Selleck Chemicals, USA). aPD-1 antibody was purchased from BioXCell (USA; Cat# BE0146), and anti-SAA1/2 was purchased from R&D Systems (USA; Cat# AF2948). Napabucasin (10 mg/kg) and lenvatinib (10 mg/kg) were given by intraperitoneal injection and intragastric administration daily for a total of 12 days.

All the mice were dynamically performed noninvasively in vivo multispectral fluorescence imaging using the In vivo imaging system (IVIS) Spectrum (PerkinElmer, USA) with Living Image software (version 4.4). On day 18, mice were sacrificed by carbon dioxide asphyxiation, and the mice's body weights and liver weights were recorded. Infiltrating leukocytes from peritumoral liver tissue were isolated for flow cytometry analysis. All procedures were performed following the guidelines of the Animal Care and Use Committee of SYSUCC.

## Hydrodynamic injection HCC model

In the hydrodynamic injection HCC model, hydrodynamic gene transfer was accomplished by rapidly injecting a solution containing gene constructs through the tail vein of C57BL/6 J mice. The three gene constructs encoding included (1) human *c-myc*, luciferase, (2) a gRNA guide for *p53* and the CAS9 nuclease, (3) a sleeping beauty transposase expression cassette. After 21 days of injection, all the mice underwent noninvasive in vivo multispectral fluorescence imaging using the in vivo imaging system (IVIS) Spectrum (PerkinElmer, USA) with Living Image software (version 4.4) to determine tumor number and extent. The mice were randomly assigned to three treatment groups and one control based on fluorescence values (*n* = 4 for each group): (1) intraperitoneal injections of vehicle (PBS with 5% DMSO); (2) intraperitoneal injections of aPD-1 antibody (10 mg/kg; dilution: 1 mg/mL); (3) intraperitoneal injections of anti-SAA1/2 antibody (5 μg per mouse; dilution: 25 μg/mL); and (4) intraperitoneal injections of aPD-1 antibody (10 mg/kg) and anti-SAA1/2 antibody (5 μg per mouse) per three days for a total of three times. After each administration, all mice underwent dynamic multispectral fluorescence imaging to monitor changes in fluorescence signals. Three days after the last administration, the mice were sacrificed by carbon dioxide asphyxiation, and necropsied were conducted. Body weights and liver weights of the mice were recorded. Formalin-fixed paraffin-embedded liver sections containing tumor and peritumor tissue were prepared for multiplex immunofluorescence staining.

## TCGA data collection and analysis

For The Cancer Genome Atlas (TCGA) cohort, clinical and RNA-seq data related to 424 HCC samples (including 50 paired tumor and peritumor normal tissues) were collected from the FireBrowse data portal. We used the single-sample gene set enrichment analysis (ssGSEA) algorithm implemented in the R package's gene set variation analysis (GSVA) to calculate the enrichment scores of immune cell type infiltration. Spearman's correlation was applied to quantify the association between SAA expression and immune cell infiltration enrichment scores. The samples were stratified into two groups according to their SAA expression (low or high), and the 50th percentiles were defined as cutoff thresholds.

## Statistical analysis

The obtained data is the results of at least two independent trials. Unless otherwise specified, data are reported as the mean ± standard error of the mean (SEM).

The statistical significance of means between two groups was calculated using a two-tailed Student's *t* test. Statistical significance is evaluated by one-way analysis of variance (ANOVA) for comparisons between more than two groups.

Receiver-operating characteristic (ROC) analyses are used to determine the cut-off value of biomarkers for distinguishing between cases with non-progressive diseases (NPD) and progressive diseases

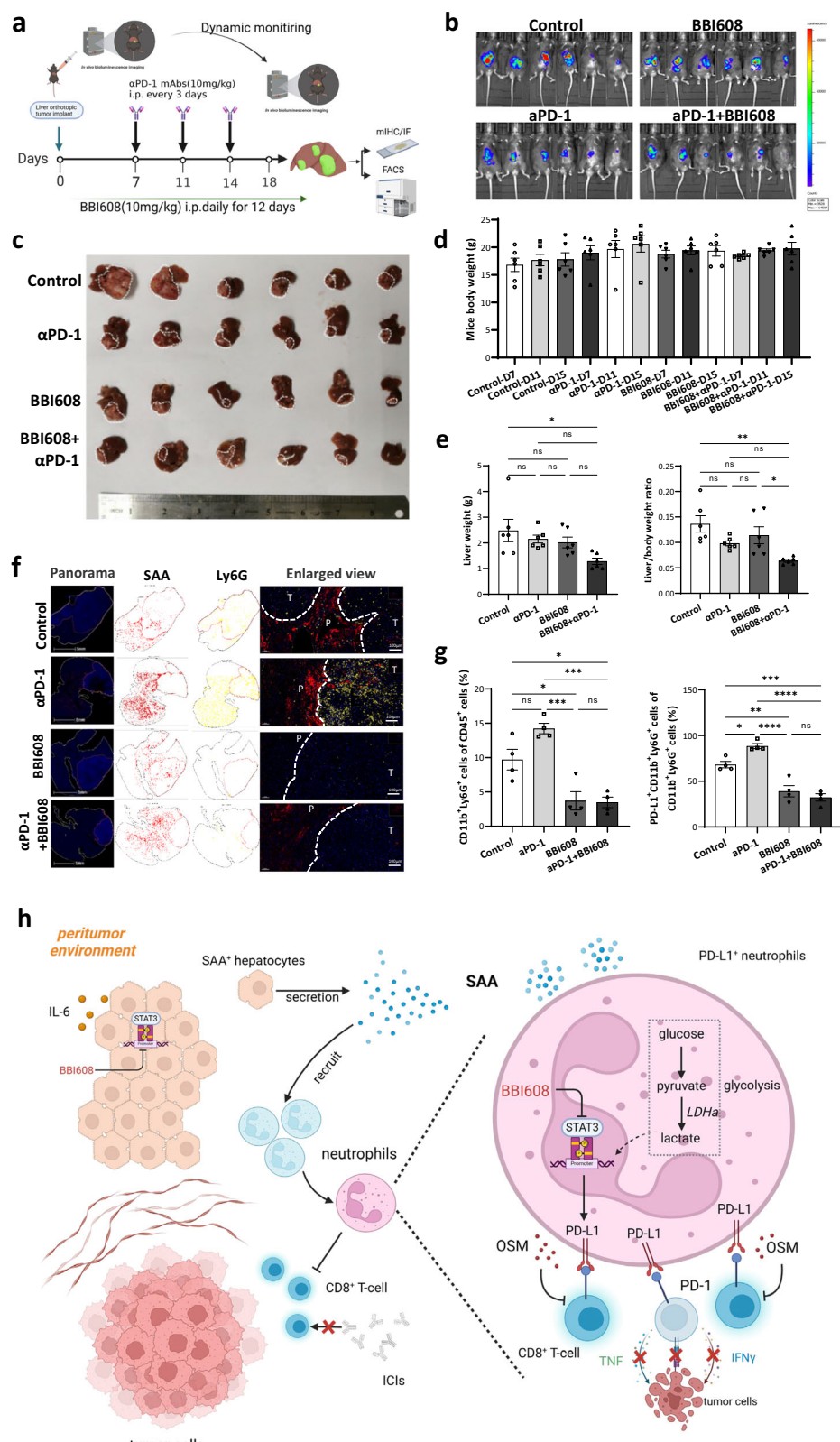

(PD). For survival comparison, differences in Kaplan-Meier curves between the groups are assessed by log-rank tests. The Cox model evaluates prognostic factors using univariate and multivariate analysis. A binary logistic regression and LASSO multivariate analysis are utilized to find tumor response predictors. All statistical analyses were performed using GraphPad Prism V9.0 (GraphPad Software, San

Diego, CA) and SPSS 26.0 (IBM Corp., Armonk, NY). The differences are considered statistically significant at *P* value < 0.05.

## Reporting summary

Further information on research design is available in the Nature Portfolio Reporting Summary linked to this article.

**Fig. 8 | αPD-1 shows synthetic anti-HCC activity in combination with STAT3 inhibitor in vivo.** Orthotopic liver murine models implanted with Hepa1-6-luci⁺ cells were separately treated with PBS combined with DMSO, αPD-1 (10 mg/kg), napabucasin/BBI608 (10 mg/kg), or αPD-1 (10 mg/kg) combined with napabucasin/BBI608 (10 mg/kg) per three days. $n = 6$ mice. After 18 days, the mice were sacrificed and necropsied. The whole body and organs of mice were weighed after treatment. The dissected tumor tissues were prepared for histological examination, FACS, and immunofluorescence analysis. **b** Before sacrifice, luciferase marker expression was detected using IVIS. $n = 6$ mice. **c** Images of tumor tissue samples were taken from the necropsied orthotopic mice model. $n = 6$ mice. **d**, **e** Histograms showed weight changes in the whole mouse, liver tumor, and liver to mouse ratio in different treatment groups. $n = 6$ mice. **f** Multiplexed immunofluorescence images showed the protein expression and location of SAA and Ly6G in each group. **g** Tumor infiltration of neutrophils (CD11b⁺Ly6G⁺ cells) and PD-L1⁺ neutrophils (PD-L1⁺CD11b⁺Ly6G⁺ cells) were analyzed by FACS in each group. $n = 4$ mice. **h** The scheme of the underlying mechanism. The results are expressed as the mean ± SEM. Statistical data presented in this figure show mean ± SEM. ns indicates $P > 0.05$, $*P < 0.05$, $**P < 0.01$, $***P < 0.001$, and $****P < 0.001$, by one-way ANOVA (**e**, **g**). Source data and exact $P$ values are provided as a Source Data file. Illustrations created with BioRender.com.

## Data availability
All study data are presented in the manuscript and supplementary materials. All the relevant raw data that support the findings of this study have been deposited in the Research Data Deposit public platform under the accession number RDDB2023340106. The raw sequence data reported in this paper have been deposited in the Genome Sequence Archive under the accession number HRA005809. Source data are provided with this paper.

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

## Acknowledgements

We thank Drs. Min-Shan Chen and Peng Huang for valuable suggestions. A portion of this research was supported by the National Natural Science Foundation of China (No. 81901850, Ning Lyu; No. 82072022 and No. 81771956, Ming Zhao) and the Guangdong Basic and Applied Basic Research Foundation (NO. 2021A1515010403, Ning Lyu).

## Author contributions

N.L., M.Z., and M.H. contributed to study conception and design. M.H., Y.X.L., C.F., S.Q., S.C., Y.H., H.M.C., X.W., and X.Y.J. contributed to collection and assembly of data. M.H., Y.X.L., H.J.D., S.Q., S.C., Q.F.C. and N.L. contributed to data analysis and interpretation. X.J.X. and M.Z. contributed to study supervision. N.L., H.J.D., M.H., Y.X.L., S.C. and M.Z. contributed to manuscript drafting. N.L., M.Z, Y.X.L. and S.C. contributed to revise the paper. All authors contributed toward final approval of manuscript and accountable for all aspect of the work.

## Competing interests

The authors declare no competing interests.
