## [Peer Review File · Nature Communications]

Serum amyloid A promotes glycolysis of neutrophils during PD-1 blockade resistance in hepatocellular carcinomaEditorial Note: Parts of this Peer Review File have been redacted as indicated to maintain the confidentiality of unpublished data.

REVIEWER COMMENTS

Reviewer #1 (Remarks to the Author):

In this study the authors explore predictive markers of response to anti-PD-1 therapies in hepatocellular carcinoma (HCC). In this retrospective study 569 patients received anti-PD-1 therapy. 243 were excluded while the study population, 326 patients', received anti-PD-1 alone or in combo with tyrosine kinase inhibitors. 52 of those just receiving anti-PD-1 were analyzed for 16 biomarkers while 274 patients with combination therapy were examined for serum amyloid A (SAA) concentration – control group. Outcomes were overall survival and radiologic response to therapy (RECIST). SAA and PLR serum markers had an association with poorer median overall survival while serum SAA, CRP, NLR and PLR predicted response (progressive disease PD vs nonprogressive disease NPD).

The first portion of the studies show RNA transcriptome analysis of 6 patients' tumor and peritumor tissue separated NPD and PD with poorer outcome having higher inflammatory markers Figure 1B and C. 17 additional HCC patients were used to validate the serum SAA link between circulating levels and poor outcome. Figure 1J.

The next portion of the studies attempt to link SAA to STAT3 signaling in peritumor tissue. This is done with data from the TCGA, tissue staining and western blots. Also linking SAA to tumor infiltrating immune cells, particularly neutrophils in HCC. The neutrophils appear to be PD-L1 expressing and correlate with a poor prognosis in PD compared to NPD cases.

The last portion of the study further dives into the STAT3 -> SAA -> (glycolysis) PD-L1 pathway with in vitro, orthotopic HCC murine model and development of genetically deficient (Saa1 and Stat3) mice.

The authors conclude that SAA is an inflammatory cytokine that predicts resistance to anti-PD-1 therapy. SAA causes neutrophils to expressing PD-L1 through STAT3 activation.

Major

Figures are too small, not legible and poorly constructed. For example, Figure 1A panels are tiny, 1C is needlessly large, 1D is not evaluable and 1G is not evaluable. This would usually

be a minor concern but THIS IS SO SIGNIFICANT IT PRECLUDES EVALUATION OF THE STUDY.

This is a remarkable series and number of technical experiments. However, there is lack of focus in the central story. The studies should be considerably streamlined.

The most important conclusion, SAA is a predictive biomarker, needs substantial validation with an independent large cohort of patients.

Minor

Slight confusion of the terms PROGNOSTIC and PREDICTIVE biomarkers. Example line 155 “predicting survival outcomes”. Prognostic is outcome and predictive is response.

Reviewer #2 (Remarks to the Author):

Very impressive work with a very nice story on a very hot topic for defining the good candidates for the best treatment for advanced HCC patients.

However, the paper suffer from several major limitations:

- Patient’s cohorts must be clearly described (4 different cohorts and only a small part of the first one is described):

o Which anti-PD1 was used in your 52 patients? What is the combination with TKI and which TKI was used and why? To my knowledge, there is currently no approval of TKI + anti-PD1!

o In suppl mat and meth, “among the 243” should be “among the remaining 326 patients”

o Suppl fig 1 is not about 274 patients but only on 52 patients (have to be corrected page 6), moreover, in this figure, size and number of HCC are not described, and finally, how many patients are cirrhotics?

o What are the characteristics of the second cohort (from 2020 and 2022) for fresh blood?

o What are the characteristics of the third cohort of 160 patients receiving and PD-1 and resected? Treatment was neo-adj or adjuvant?

o Finally, what are the characteristics of the forth cohort with snap-freezed fresh core-needle biopsy for RNA-seq? How are you sure that the nodules are HCC?

- Statistical analysis have to be stronger.

o Indeed, 16 variables are tested on only 52 patients. To improve the strength of the results, LASSO multivariate analysis have to be done in order to avoid a result that could come from hazard.

o Definition of responders have to be stronger. In the present work, the definition include

CR, PR and SD. Response to treatment means decrease of the size of the tumor. Objective response is internationally recognized as CR + PR only. Patients with SD must be considered as non-responders.

- Cell lines:

o Human Hepatocyte cell line used in this work (Lo2) is no more considered as a liver cell line but a cercical cell line. Data have to be confirmed with several cell lines!

o Murine cell line of HCC used in the experiments is Hep1-6. This cell lines is very sensitive to IO treatment. Mice experiments must be confirmed with immuno-negative cell lines such as the one coming from a TP53 or a β catenin background.

Reviewer #3 (Remarks to the Author):

In their manuscript, the authors provide evidence, in advanced HCC, for a higher serum amyloid A (SAA) expression levels in the peritumoral tissue upon anti-PD1 treatment in progressive disease (PD) compared to non-progressive disease (NPD) patients. In the circulation, SAA levels correlate with peritumoral SAA expression, and increase when PD occurs. The authors suggest that IL6 promotes SAA induction in a STAT3-dependent manner in peritumoral hepatocytes. In peritumoral tissue, SAA correlated with neutrophil and PDL1+ neutrophil abundance. In vitro, SAA induced PDL1 in neutrophils, and the expression of glycolytic genes. SAA-stimulated PDL1 was blocked upon 2-deoxyglucose usage or LDHA inhibition. Functionally, SAA treatment of neutrophils suppressed IFN γ and TNF production by autologous CD8 T cells. Finally, using a mouse orthotopic liver tumor model, SAA blockade synergised with anti-PD1, and SAA gene deletion enabled the response of liver tumor cells to anti-PD1. Similarly, Stat3 gene deletion in the liver epithelial cells sensitised tumors to anti-PD1. Of note, each of SAA and Stat3 deletion were done in the recipient mice, not in the tumor cells, showing the importance of the non-tumor tissue in the observed phenomena. Stat3 pharmacological blockade recapitulated the effect of its gene deletion in terms of sensitisation to anti-PD1.

I have a few specific comments, which I hope will help improve the current manuscript:

Major comments:

1- Only a single orthotopic tumor cell line injection model is used. Could authors use another model, at least another cell line injected orthotopically into immunocompetent mice? Otherwise, there is a risk that some findings are specific to this cell line.

2- In Fig. 8B and Fig. 10B, there is no information on the bioluminescence values before treatment. Do the authors have this information? Could a graphic be added, showing the bioluminescence value for each mouse over time, with associated statistical analyses?

3- There is no quantification of the percentage of colocalization (only a visual observation by mIHC). Can authors add a graphic with percentage of colocalization for mIHC experiments (fig.2 C fig.2E, fig.2.F, fig.4E, fig.7E, fig.9H, fig.9I)? (For example, for fig. 4E: levels of CD15+PD-L1+ cells and CD15+PD-L1+pSTAT3+ cells should be shown in the tumor or in the peritumour region).

4- In figure 7D, the decrease of OSM in SAA+2-DG condition compared to SAA condition is not significant (the contrary is stated in the text page 13). Please explain. Also, because it is the only experiment monitoring glycolytic involvement in OSM secretion, could the authors make a stronger demonstration involving glycolysis in OSM secretion?

5- It is important to show the link between OSM secretion by SAA-stimulated neutrophils and CD8 T cell activation. Could the authors make an experiment in which neutrophils are treated with SAA or are treated with SAA and several doses of an OSM inhibitor, and monitor CD8 T cell activation (by flow cytometry as in Fig. 7F)?

Minor comments:

1- in vitro experiments have been performed with blood-derived neutrophils from HCC patients. If taken from healthy donor, do neutrophils also increase expression of PDL1, glycolytic genes and become more glycolytic in response to SAA?

2- Is SAA-mediated OSM upregulation also inhibited upon LDHA blockade?

3- Could authors choose images in Fig. 3A that are representative of their quantifications (Fig. 3B)? Visually, the important infiltration of CD8 T cells in SAA low samples is confusing

with quantifications (showing an increase of CD8 T cells infiltration in SAA high samples).

4- Could authors check for the presence of glycolytic+PD-L1+CD15+pSTAT3+ cells in peritumour/tumor? And their physical proximity with CD8 T cells?

5- In the context of Stat3 deficiency, the authors stated that IFNg+ cells are T cells. Could authors add a CD3 or CD8 marker to the mIHC panel? (Fig. 9I).

6- For the flow cytometry data, can the authors indicate the percentage of what they refer to on the y axis?

Editing comments:

- In Fig. 2F we cannot see well the difference between stainings (red and orange). Maybe pSTAT3 representation could be changed to green?

- Some graphics are too small (for example Fig. 1A).

- In Fig. 2F, most arrows indicate empty spaces. Could the authors readjust this?

- In Fig. 8G, images of the aSAA1/2 + aPD-1 condition are all shown with their corresponding channel + the red channel. Red channel should be deleted from all other channels except for the red and merge ones.

- In suppl. Fig. 8 author should indicate if the green area represents tumor or peritumour.

Also, it seems that PD-L1+ cells are indicated in black, and not PD-L1 neutrophils (the name PD-L1 neutrophils should be changed to PD-L1 cells).

- In Fig. 8G and Fig. 9, the tumor and the peritumour region should be indicated.

- Molecular weight of observed bands should be added for Fig. 4C and Fig.4D.

- In Fig. 4F, the lines between each statistical test should be readjusted.

- Fig. 5E and Fig. 6A are the only graphs not represented with points (showing each data).

These graphs should be adjusted.

- Page 13, second line, "Supplementary fig. 11C" should be changed for "Supplementary fig. 10C".

- Page 13, "MPO+OSM+ are mostly in peritumour" could be changed to invasion margin.

- Last paragraph of page 5, "And PD-L1+ neutrophils [...] immune escape" should be corrected.

- Page 8 “Moreover, the pre-treatment cytoplasm protein expression of SAA” should be corrected.

We appreciate the helpful comments from the reviewers and revised the manuscript accordingly to clarify our finding. The detailed responses and quotations from the revised manuscript are listed below.

Reviewer #1' comments:

In this study the authors explore predictive markers of response to anti-PD-1 therapies in hepatocellular carcinoma (HCC). In this retrospective study 569 patients received anti-PD-1 therapy. 243 were excluded while the study population, 326 patients', received anti-PD-1 alone or in combo with tyrosine kinase inhibitors. 52 of those just receiving anti-PD-1 were analyzed for 16 biomarkers while 274 patients with combination therapy were examined for serum amyloid A (SAA) concentration – control group. Outcomes were overall survival and radiologic response to therapy (RECIST). SAA and PLR serum markers had an association with poorer median overall survival while serum SAA, CRP, NLR and PLR predicted response (progressive disease PD vs nonprogressive disease NPD).

The first portion of the studies show RNA transcriptome analysis of 6 patients' tumor and peritumor tissue separated NPD and PD with poorer outcome having higher inflammatory markers Figure 1B and C. 17 additional HCC patients were used to validate the serum SAA link between circulating levels and poor outcome. Figure 1J.

The next portion of the studies attempt to link SAA to STAT3 signaling in peritumor tissue. This is done with data from the TCGA, tissue staining and western blots. Also linking SAA to tumor infiltrating immune cells, particularly neutrophils in HCC. The neutrophils appear to be PD-L1 expressing and correlate with a poor prognosis in PD compared to NPD cases.

The last portion of the study further dives into the STAT3 -> SAA -> (glycolysis) PD-L1 pathway with in vitro, orthotopic HCC murine model and development of genetically deficient (Saa1 and Stat3) mice.

The authors conclude that SAA is an inflammatory cytokine that predicts resistance to

anti-PD-1 therapy. SAA causes neutrophils to expressing PD-L1 through STAT3 activation.

Major

1. Figures are too small, not legible and poorly constructed. For example, Figure 1A panels are tiny, 1C is needlessly large, 1D is not evaluable and 1G is not evaluable. This would usually be a minor concern but **THIS IS SO SIGNIFICANT IT PRECLUDES EVALUATION OF THE STUDY.**

Response: Thank you for the valuable and encouraging comment. We sincerely apologize for any inconvenience caused by the pre-matured figure construction in the previous version of the manuscript. Following your suggestion, we have re-constructed the figures and enlarged the panels to make sure they are clear and easy for reading.

- **Revised Fig. 1**

[**Editorial Note:** Figure redacted]

2. This is a remarkable series and number of technical experiments. However, there is lack of focus in the central story. The studies should be considerably streamlined.

Response: We fully agreed with the Reviewer's comment regarding the lack of focus among the series and number of technical experiments in the previous version of our manuscript. Taking into account the initial and additional experiments as well as clinical data, we have reorganized and adjusted the entire contents to emphasize the central story and ensure a streamlined approach to the study.

The main objective of our study was to investigate the role of serum amyloid A (SAA) levels in advanced HCC patients undergoing anti-PD1 treatment. We observed higher SAA expression in the peritumoral tissue of progressive disease (PD) patients compared to nonprogressive disease (NPD) patients. Additionally, circulating SAA levels correlated with peritumoral SAA expression and increased during PD.

Further analysis revealed that SAA expression in peritumoral tissue correlated with neutrophil abundance and the presence of PD-L1+ neutrophils. *In vitro* experiments demonstrated that SAA induced upregulation of PD-L1 in neutrophils and promoted the overexpression of glycolytic genes, both could be blocked by 2-deoxyglucose or LDHA inhibition.

Functionally, SAA-treated neutrophils suppressed the production of IFN γ and TNF α by autologous CD8+ T cells. To validate these findings *in vivo*, we utilized orthotopic liver tumor murine models and a hydrodynamic injection model. The blockade of SAA or genetic deletion of SAA synergized with anti-PD-1 treatment on liver tumor models. Similarly, *Stat3* gene deletion in the liver epithelial cells sensitized tumors to anti-PD-1 treatment. Importantly, both SAA and *Stat3* deletions were performed in recipient mice rather than tumor cells, highlighting the significance of non-tumor tissue in the observed phenomena. Pharmacological blockade of STAT3 recapitulated the effect observed with its gene deletion, enhancing sensitivity to anti-PD-1 treatment.

In summary, our study focused on the predictive and prognostic role of SAA levels in both circulation and peritumoral tissue in resistance to PD-1 blockade in hepatocellular carcinoma. We demonstrated that SAA facilitated neutrophil glycolysis to mediate the PD-1 blockade resistance. Therefore, combining SAA or STAT3 inhibitors with anti-PD-1 therapy may be a promising approach to enhance its efficacy.

- **Revised title**

“Serum amyloid A induces glycolysis of neutrophils for PD-1 blockade resistance in hepatocellular carcinoma”

- **Revised results**

- 1. SAA correlates with the efficacy of PD-1 blockade in aHCC patients
- 2. Peritumoral SAA-related PD-L1+ neutrophils affect anti-PD-1 response
- 3. SAA induces PD-L1 upregulation of neutrophils via STAT3 pathway
- 4. SAA induces PD-L1 expression through glycolytic activation
- 5. SAA-treated neutrophils increase OSM to inhibit CD8+ T cells activity
- 6. SAA blockade or *Saa1* ablation enhances anti-PD-1 efficacy *in vivo*

- 7. STAT3 inhibition enhances efficacy of anti-PD-1 based therapy in vivo

3. The most important conclusion, SAA is a predictive biomarker, needs substantial validation with an independent large cohort of patients.

Response: Thank you for the valuable comment and suggestion. It is indeed necessary to validate with an independent cohort to support the conclusion that SAA is a predictive biomarker for advanced HCC patients receiving first-line anti-PD-1 based treatment. In the revised manuscript, we retrospectively collected 138 patients with advanced HCC receiving anti-PD-1 based treatment from Jan, 2021 to Jan, 2023. The baseline characteristics of the 138 patients were listed in **Supplementary Table 5**. As a result, the statistical analysis of the validation cohort demonstrated that SAA had a parallel effect on predicting tumor responses and prognosing survival outcomes in advanced HCC patients receiving anti-PD-1 based treatment, consistent with the original cohort of 52 patients treated with anti-PD-1 monotherapy (**revised results: page 7, line 135–138; Supplementary Fig. 4f and 4g**). The corresponding results and figure were shown as follows:

- **Revised Supplementary Fig. 4f and 4g**

[**Editorial Note:** Figure redacted]

Supplementary Fig. 4. Pre-treatment circulating SAA (cutoff = 20 mg/L) is a potential predicting factor and prognostic factor of survival. (f-g) In the validation cohort of 138 aHCC patients treated with PD-1 inhibitor-based combination treatment.

Minor

1. Slight confusion of the terms PROGNOSTIC and PREDICTIVE biomarkers. Example line 155 “predicting survival outcomes”. Prognostic is outcome and predictive is response.

Response: Thanks for the comment. We apologize for any confusion caused by our incorrect description. We appreciate your suggestion and have made the necessary corrections (**revised results: page 5, line 94; page 6, line 122–123**). Furthermore, we have conducted a thorough review of the entire manuscript to identify and correct any similar confusing descriptions.

Reviewer #2' comments:

Very impressive work with a very nice story on a very hot topic for defining the good candidates for the best treatment for advanced HCC patients. However, the paper suffers from several major limitations:

1 Patient's cohorts must be clearly described (4 different cohorts and only a small part of the first one is described):

1.1 Which anti-PD1 was used in your 52 patients? What is the combination with TKI and which TKI was used and why? To my knowledge, there is currently no approval of TKI + anti-PD1!

Response: Thank you for the comment. We apologize for the unclear descriptions of the cohorts. The anti-PD-1 drugs used in the 52 patients included pembrolizumab, nivolumab, sintilimab and toripalimab. The distribution of patients treated with each specific anti-PD-1 category is provided in revised **Supplementary Table 1**. The 274 patients who received combined therapies were divided into three categories: anti-PD-1 combined with TKIs, anti-PD-1 combined with locoregional therapy, and anti-PD-1 combined with TKIs and locoregional therapy. The specific categories of TKIs used in these 274 patients included lenvatinib, sorafenib and regorafenib. The number of patients treated with each TKI category is provided in the **Supplementary Table 4**.

Furthermore, in response to the question of "*What is the combination with TKI and which TKI was used and why? To my knowledge, there is currently no approval of TKI + anti-PD1!*", we acknowledge that there is currently no definite approval for this combination in global guidelines such as BCLC, EASL, and AASLD (1-3). However, during the course of our study from 2017 until now, several phase 1 and 2 clinical trials have been conducted exploring the combination of TKIs and anti-PD-1 for aHCC patients. Some of these trials have reported improved outcomes, such as lenvatinib plus pembrolizumab, apatinib plus camrelizumab, and anlotinib plus penpulimab (4-6).

Based on these results, the combination of TKIs and anti-PD-1 has been recommended as one of the first-line regimens for aHCC patients in **Chinese guidelines** since 2022 and has been included in the coverage of medical health insurance in China (7, 8). In addition, although the phase 3 clinical trial of lenvatinib and pembrolizumab (LEAP-002) did not show a significant improvement in survival compared to lenvatinib monotherapy for aHCC patients globally, it did demonstrate clinical activity and a favorable safety profile, indicating improvements in tumor responses and an enhanced antitumor response relative to TKIs (9, 10). Furthermore, another phase 3 clinical trial of apatinib combined with camrelizumab (CARES-310) demonstrated improved outcomes compared to sorafenib in 2023, including overall survival, progression-free survival, and tumor response (11). Therefore, the combination of TKIs and anti-PD-1 has been widely used in clinical practice in China and has shown superior outcomes compared to TKIs monotherapy, with a relatively safe profile (12).

References:

1. Reig M, Forner A, Rimola J, Ferrer-Fabrega J, Burrel M, Garcia-Criado A, Kelley RK, et al. BCLC strategy for prognosis prediction and treatment recommendation: The 2022 update. *J Hepatol* 2022;76:681-693.
2. European Association for the Study of the Liver. Electronic address eee, European Association for the Study of the L. EASL Clinical Practice Guidelines: Management of hepatocellular carcinoma. *J Hepatol* 2018;69:182-236.
3. Heimbach JK, Kulik LM, Finn RS, Sirlin CB, Abecassis MM, Roberts LR, Zhu AX, et al. AASLD guidelines for the treatment of hepatocellular carcinoma. *Hepatology* 2018;67:358-380.
4. Richard S. Finn, Masafumi Ikeda, Andrew X. Zhu, Andrew X. Zhu, et al. Phase Ib Study of Lenvatinib Plus Pembrolizumab in Patients With Unresectable Hepatocellular Carcinoma. *J Clin Oncol* 2020;38:2960-2970.
5. Xu J, Shen J, Gu S, Zhang Y, Wu L, Wu J, Shao G, et al. Camrelizumab in Combination with Apatinib in Patients with Advanced Hepatocellular Carcinoma (RESCUE): A Nonrandomized, Open-label, Phase II Trial. *Clin Cancer Res* 2021;27:1003-1011.
6. Han C, Ye S, Hu C, Shen L, Qin Q, Bai Y, Yang S, et al. Clinical Activity and Safety of Penpulimab (Anti-PD-1) With Anlotinib as First-Line Therapy for Unresectable Hepatocellular Carcinoma: An Open-Label, Multicenter, Phase Ib/II Trial (AK105-203). *Front Oncol* 2021;11:684867.
7. Administration NHCotPsRoCM. Consensus Statement Guidelines for Diagnosis and Treatment of Primary Liver Cancer in China (2022 Edition). *Chin J Clin Hepatol* 2022;38:288-303.

8. (CSCO) CSOCo. Guidelines of Chinese Society of Clinical Oncology (CSCO): hepatocellular carcinoma (2022). 2022.
9. Llovet JM, Castet F, Heikenwalder M, Maini MK, Mazzaferro V, Pinato DJ, Pikarsky E, et al. Immunotherapies for hepatocellular carcinoma. *Nat Rev Clin Oncol* 2022;19:151-172.
10. Yi C, Chen L, Lin Z, Liu L, Shao W, Zhang R, Lin J, et al. Lenvatinib Targets FGF Receptor 4 to Enhance Antitumor Immune Response of Anti-Programmed Cell Death-1 in HCC. *Hepatology* 2021;74:2544-2560.
11. Qin S, Chan SL, Gu S, Bai Y, Ren Z, Lin X, Chen Z, et al. Camrelizumab plus rivoceranib versus sorafenib as first-line therapy for unresectable hepatocellular carcinoma (CARES-310): a randomised, open-label, international phase 3 study. *Lancet* 2023.
12. Yang X, Chen B, Wang Y, Wang Y, Long J, Zhang N, Xue J, et al. Real-world efficacy and prognostic factors of lenvatinib plus PD-1 inhibitors in 378 unresectable hepatocellular carcinoma patients. *Hepatol Int* 2023;17:709-719.

1.2 In suppl mat and meth, “among the 243” should be “among the remaining 326 patients”

Response: Thank you for pointing out this mistake, and we have corrected it in the revised manuscript (**revised methods: page 18, line 437**).

1.3 Suppl fig 1 is not about 274 patients but only on 52 patients (have to be corrected page 6), moreover, in this figure, size and number of HCC are not described, and finally, how many patients are cirrhotic?

Response: Thank you for your comment. We apologize for the improper arrangement of the flow chart in the original Supplementary Fig. 1A, which made it difficult to understand. In fact, the original intention for Supplementary Fig. 1A was to include both the 274 patients and 52 patients, rather than just the 52 patients. We have taken this into account and re-drew the flowchart to ensure clarity and accuracy (**revised Supplementary Fig. 1a**). Besides, we have included the size and number of HCC cases, as well as the proportion of cirrhotic patients in **Supplementary Table 1 and 4**.

- **Original Supplementary Fig. 1A**

[Editorial Note: Figure redacted]

- **Revised Supplementary Fig. 1a**

[Editorial Note: Figure redacted]

Supplementary Fig. 1. Study population screening and preliminary assessment of the prognostic value of SAA in patients treated with anti-PD-1 immunotherapy. (a) Flowchart of the study population screening.

1.4 What are the characteristics of the second cohort (from 2020 and 2022) for fresh blood?

Response: Thank you for the question. We have provided the characteristics of the second cohort, consisting of 68 patients from 2020 and 2022, in terms of fresh blood analysis. The detailed data were listed in the revised **Supplementary Table 7**.

1.5 What are the characteristics of the third cohort of 160 patients receiving PD-1 and resected? Treatment was neo-adj or adjuvant?

Response: Thank you for your comment and question. We have provided the characteristics of the third cohort consisting 160 patients who underwent surgical resection. The baseline characteristics of these patients were listed in revised **Supplementary Table 6**. However, there may have been confusion due to the following statement: "*Further, formalin-fixed, paraffin-embedded samples of HCC patients receiving anti-PD-1 monotherapy and tissue microarray (TMA) of resected HCC tumors were utilized for multiplexed immunohistochemistry (mIHC) analysis.*" This statement actually refers to two different cohorts. The 160 patients received resection monotherapy and did not receive anti-PD-1 therapy before or after resection in the TMA cohort. Samples from these patients was used for mIHC analysis to demonstrate

that SAA was expressed at a higher level in peritumoral tissues than in tumor tissues. Therefore, we have revised the statement in the manuscript to provide clarity on this matter.

- **Modified materials and methods (page 17, line 419–422)**

“Further, formalin-fixed, paraffin embedded samples of HCC patients receiving anti-PD-1 monotherapy and tissue microarray (TMA) of resected HCC tumors without anti-PD-1 treatment were utilized for multiplexed immunohistochemistry (mIHC) analysis.”

1.6 Finally, what are the characteristics of the fourth cohort with snap-frozen fresh core-needle biopsy for RNA-seq? How are you sure that the nodules are HCC?

Response: Thank you for your comment. There were a total of six patients involved in the snap-frozen fresh core-needle biopsy for RNA-seq. These patients were selected from the cohort of 52 patients who received anti-PD-1 monotherapy. The baseline characteristics of these patients were listed in **Supplementary Table 3**. Besides, the nodules were firmly diagnosed as HCC by parallel pathological examination of tissue samples in the Pathology Department. We have provided this statement in the revised manuscript (**revised methods: page 18, line 426–428**).

2 Statistical analysis has to be stronger:

2.1 Indeed, 16 variables are tested on only 52 patients. To improve the strength of the results, LASSO multivariate analysis has to be done in order to avoid a result that could come from hazard.

Response: We appreciate your comment on the statistical analysis and methodology. In order to address potential biases and strengthen our results, we conducted LASSO multivariate analysis. As the result shown, consistent with the findings from the

previous COX analysis, SAA, CRP, and NLR emerged as the three potential predictive factors among the 16 markers analyzed using LASSO multivariate analysis. The corresponding methods and figure were shown as follows:

- **Revised materials and methods (The “LASSO multivariate analysis” section)**

“LASSO multivariate analysis was used to verify the predicting factors for tumor response (PD/NPD) among the 16 variables. The binomial deviance curve was plotted versus $\log(\lambda)$, where λ is a tuning hyperparameter. Solid vertical lines represent binomial deviance \pm standard error (SE). The dotted vertical lines are drawn at optimal values by using the minimum criteria and 1-SE criteria. An optimal λ value was selected for the LASSO model by using 10-fold cross-validation via minimum criteria. A coefficient profile plot was produced against the $\log(\lambda)$ sequence.”

- **Revised Supplementary Fig. 2d**

[Editorial Note: Figure redacted]

Supplementary Fig. 2 Statistical assessment of the 16 serum biomarkers for aHCC patients treated with single anti-PD-1 immunotherapy. (d) LASSO multivariate analysis further demonstrated that baseline SAA, CRP and NLR were predicting factors to differentiate PD and NPD. Ten time cross-validation for tuning parameter selection in the LASSO model (left), and the LASSO coefficient (right). The linear predictor was defined as $(-1.76384186) + \text{SAA} \times (0.0092548) + \text{CRP} \times (0.00042891) + \text{NLR} \times (0.07575887)$.

2.2 Definition of responders have to be stronger. In the present work, the definition include CR, PR and SD. Response to treatment means decrease of the size of the tumor. Objective response is internationally recognized as CR + PR only. Patients with SD must be considered as non-responders.

Response: Thanks for the comment. We agree that objective response is comprised with complete response (CR) and partial response (PR) according to RECIST 1.1 creteria. However, the response category for cases with stable disease (SD) should be divided

into two parts. One part includes cases where the sum of diameters of target lesions decreases by less than 30%, and the other part includes cases where the diameter increases less than 20% (1). This definition indicates that SD refers to cases not only have increased lesions but also have decreased lesions in part. Our study mainly focused on resistance rather response to anti-PD-1. Resistance principally refers to progressive disease (PD) after receiving treatment (2). Therefore, we used a strict criterion where patients with PD are considered resistant to anti-PD-1, while patients with no progressive disease (NPD) are considered non-resistant to anti-PD-1.

References:

1. E.A. Eisenhauer, P. Therasse , J. Bogaerts , L.H. Schwartz , D. Sargent , R. Ford, et al. New response evaluation criteria in solid tumours: Revised RECIST guideline (version 1.1). *Eur J Cancer* 2009;45:228-247.
2. Sreya Bagchi , Robert Yuan , Edgar G Engleman. Immune Checkpoint Inhibitors for the Treatment of Cancer: Clinical Impact and Mechanisms of Response and Resistance. *Annu Rev Pathol* 2021;16:223–249.

3 Cell lines:

3.1 Human Hepatocyte cell line used in this work (Lo2) is no more considered as a liver cell line but a cervical cell line. Data have to be confirmed with several cell lines!

Response: Thanks for your comment. We appreciate your insight for the clarification for the information regarding the LO2 cell line. In our revision, we replaced the LO2 cell line with another hepatocyte cell line named HHL-5, and repeated the same experiment. Consistent with previous findings, Western blot assay demonstrated a significant upregulation of both p-STAT3 and SAA1+2 expression in HHL-5 cells after stimulation with IL-6 (20 ng/mL), while such stimulating effect could be abrogated by a STAT3 inhibitor napabucasin (BBI608; 0.1 μ mol/L). The corresponding methods and figure were shown as follows:

- **Revised materials and methods (page 19, line 470–474)**

“Liver cell lines HHL-5 (Yaji Biotechnology Co., Shanghai, China) derived from human hepatocytes were authenticated by short tandem repeat typing technology and were used for in vitro experiments. The murine HCC cell lines Hepa1-6-luc+ were used for in vivo experiments. Neutrophils or T cells were isolated from the blood of patients with advanced HCC (aHCC) treated at the Sun Yat-sen University Cancer Center and healthy donors.

HHL-5 and Hepa1-6 cell lines were cultured in Dulbecco’s Modified Eagle’s Medium (Gibco, China). Neutrophils and T cells were cultured in RPMI-1640 (Gibco, China). Both media were supplemented with 10% fetal bovine serum (Hyclone, New Zealand) and 1% penicillin-streptomycin (Gibco, China). All cells were cultured at 37 °C and 5% CO₂.”

- **Revised Supplementary Fig. 7d**

[Editorial Note: Figure redacted]

Supplementary Fig. 7 The activation of STAT3 signaling induced by IL-6 results in the production of SAA in peritumoral hepatocytes. (d) Western blot shows that both p-STAT3 and SAA1+2 expression in HHL-5 cells is remarkably upregulated after treatment with IL-6 (20 ng/mL), while the stimulating effect of IL-6 on p-STAT3 and SAA1+2 can be prevented after STAT3 inhibition by napabucasin (BBI608; 0.1 μmol/L).

References:

1. Hu X, Yang T, Li C, Zhang L, Li M, Huang W, et al. Human fetal hepatocyte line, L-02, exhibits good liver function in vitro and in an acute liver failure model. *Transplant Proc.* 2013, 45:695–700.
2. Ye F, Chen C, Qin J, Liu J, Zheng C. Genetic profiling reveals an alarming rate of cross-contamination among human cell lines used in China. *FASEB J.* 2015, 29:4268–72.
3. International Cell Line Authentication Committee. Register of misidentified cell lines. <https://iclac.org/databases/cross-contaminations/>. Accessed 27 Jul 2022.

3.2 Murine cell line of HCC used in the experiments is Hep1-6. This cell line is very sensitive to IO treatment. Mice experiments must be confirmed with immuno-negative cell lines such as the one coming from a TP53 or a β catenin background.

Response: We appreciate your comment. Following your suggestion, an orthotopic liver cancer model by hydrodynamic injection with *c-myc* overexpression and *TP53* knockout plasmids in C57B/6J mice was constructed to further confirm the results as an immuno-negative murine model *in vivo* (**revised methods: page 27, line 658 to page 28, line 677**). The hydrodynamic gene transfer method involved rapid injection through the tail vein of mice with a solution containing gene constructs (1). The three gene constructs encoding included: 1) human *c-myc*, luciferase, 2) a gRNA guide for p53 and the CAS9 nuclease, and 3) a sleeping beauty transposase expression cassette (2).

After 21 days, the expression of luciferase marker was detected using IVIS to confirm tumor formation. The mice were then randomised into four groups based on bioluminescence values and intraperitoneally injected with different treatments: PBS combined with 0.5% methylcellulose, α SAA1/2 (5 μ g/mice), α PD-1 (10mg/kg), or α PD-1 (10 mg/kg) combined with α SAA1/2 (5 μ g/mice) per three days for a total of three times (n = 4 for each group).

Three days after finishing last administration, the mice were sacrificed and necropsied. During the course, one mouse from the control group and one mouse from the α SAA1/2 group died due to high tumor burden before being sacrificed. The results showed that tumor growth was slightly suppressed with anti-PD-1 monotherapy. However, a synergistic effect was observed in the combination group compared to the other three groups. Additionally, the liver weight and liver to body weight ratio significantly decreased in the mice treated with the combined regimens.

In conclusion, similar to the Hep1-6 murine model, SAA blockade can enhance anti-HCC efficacy of PD-1 inhibitors in the hydrodynamic injection murine model. The

corresponding figure were shown as follows:

● **Revised Supplementary Fig. 16**

[**Editorial Note:** Figure redacted]

Supplementary Fig. 16 α SAA1/2 enhances α PD-1 efficacy in hydrodynamic injection HCC mice models. (a) Hydrodynamic injection murine models were separately treated with PBS combined with 0.5% methylcellulose, α PD-1 (10 mg/kg), α SAA1/2 (5 μ g/mice), or α PD-1 (10 mg/kg) combined with α SAA1/2 (5 μ g/mice) per three days (n = 4 for each group). After 31 days, the mice were sacrificed and necropsied. The whole body and organs of mice were weighed after treatment. (b) The three gene constructs encoding injected in the hydrodynamic injection model. (c) Images of tumor tissue samples were taken from the necropsied orthotopic mice model. (d) Prior to treatment, during the 2nd treatment, and at harvest, luciferase marker expression was dynamically detected using IVIS. (e) The confocal microscopy images showed the expression of SAA, PD-L1, MPO+ neutrophils and CD8+ T cells in each treatment group by immunofluorescence analysis. (f) Histograms showed weight changes in the liver weight and liver to body weight ratio in different treatment groups. (g) The bioluminescence value for each mouse was measured over time, and associated statistical analyses. (h) The counts of MPO+ cells, PD-L1+ MPO+ cells and CD8+ T cells in peritumor region were analyzed by mIHC in each group (n=3/group). The results are expressed as the mean \pm SEM. Statistical analyses were performed by using unpaired t-test with Mann–Whitney U. ns, no significance, *P <0.05, **P <0.01, ***P <0.001, and ****P <0.0001.

References:

1. Liu F, Song Y, and Liu D. Hydrodynamics-based transfection in animals by systemic administration of plasmid DNA. *Gene Ther* 1999;6:1258– 1266.
2. Maria Carmen Ochoa, Sandra Sanchez-Gregorio, Carlos E. de Andrea, Saray Garasa, Maite Alvarez, Irene Olivera, Javier Glez-Vaz, et al. Synergistic effects of combined immunotherapy strategies in a model of multifocal hepatocellular carcinoma. *Cell Reports Medicine* 2023;4:101009.

Reviewer #3's comment:

In their manuscript, the authors provide evidence, in advanced HCC, for a higher serum amyloid A (SAA) expression levels in the peritumoral tissue upon anti-PD1 treatment in progressive disease (PD) compared to nonprogressive disease (NPD) patients. In the circulation, SAA levels correlate with peritumoral SAA expression, and increase when PD occurs. The authors suggest that IL6 promotes SAA induction in a STAT3-dependent manner in peritumoral hepatocytes. In peritumoral tissue, SAA correlated with neutrophil and PDL1+ neutrophil abundance. In vitro, SAA induced PDL1 in neutrophils, and the expression of glycolytic genes. SAA-stimulated PDL1 was blocked upon 2-deoxyglucose usage or LDHA inhibition. Functionally, SAA treatment of neutrophils suppressed IFN γ and TNF production by autologous CD8 T cells. Finally, using a mouse orthotopic liver tumor model, SAA blockade synergized with anti-PD1, and SAA gene deletion enabled the response of liver tumor cells to anti-PD1. Similarly, Stat3 gene deletion in the liver epithelial cells sensitized tumors to anti-PD1. Of note, each of SAA and Stat3 deletion were done in the recipient mice, not in the tumor cells, showing the importance of the non-tumor tissue in the observed phenomena. Stat3 pharmacological blockade recapitulated the effect of its gene deletion in terms of sensitization to anti-PD1.

I have a few specific comments, which I hope will help improve the current manuscript:

Major

- 1 Only a single orthotopic tumor cell line injection model is used. Could authors use another model, at least another cell line injected orthotopically into immunocompetent mice? Otherwise, there is a risk that some findings are specific to this cell line.

Response: Thank you for your valuable comment, which is also pointed out by the Reviewer #2. In the question 3.2 Following your suggestion, a hydrodynamic injection

model with *c-myc* overexpression and *TP53* knockout background in C57B/6J mice was constructed to further confirm the results as an immuno-negative murine model *in vivo* (**revised methods: page 27, line 658 to page 28, line 677**). The hydrodynamic gene transfer method involved rapid injection through the tail vein of mice with a solution containing gene constructs (1). The three gene constructs encoding included: 1) human *c-myc*, luciferase, 2) a gRNA guide for p53 and the CAS9 nuclease, and 3) a sleeping beauty transposase expression cassette (2). The experimental results showed that similar to the Hep1-6 murine model, SAA blockade can enhance anti-HCC efficacy of PD-1 inhibitors in the hydrodynamic injection murine model (**revised Supplementary Fig. 16**).

References:

1. Liu F, Song Y, and Liu D. Hydrodynamics-based transfection in animals by systemic administration of plasmid DNA. *Gene Ther* 1999;6:1258– 1266.
2. Maria Carmen Ochoa, Sandra Sanchez-Gregorio, Carlos E. de Andrea, Saray Garasa, Maite Alvarez, Irene Olivera, Javier Glez-Vaz, et al. Synergistic effects of combined immunotherapy strategies in a model of multifocal hepatocellular carcinoma. *Cell Reports Medicine* 2023;4:101009.

2 In Fig. 8B and Fig. 10B, there is no information on the bioluminescence values before treatment. Do the authors have this information? Could a graphic be added, showing the bioluminescence value for each mouse over time, with associated statistical analyses?

Response: Thanks for your comment and suggestion. We apologize for not including the bioluminescence values of the mice before treatment in Fig. 8B and Fig. 10B. In those experiments, to minimize selection bias, these mice were initially randomized to different groups, and we only performed multispectral fluorescence imaging before sacrificed the mice. Therefore, we regretfully do not have the bioluminescence value for each mouse over time. In the revised work, to show the dynamic change in bioluminescence value, we conducted dynamical multispectral fluorescence imaging

for each mouse over time in the new hydrodynamic injection murine HCC model. Furthermore, we performed statistical analyses to highlight the differences (**revised Supplementary Fig. 16**).

3 There is no quantification of the percentage of colocalization (only a visual observation by mIHC). Can authors add a graphic with percentage of colocalization for mIHC experiments (fig.2C fig.2E, fig.2F, fig.4E, fig.7E, fig.9H, fig.9I)? (For example, for fig. 4E: levels of CD15+PD-L1+ cells and CD15+PD-L1+pSTAT3+ cells should be shown in the tumor or in the peritumor region).

Response: Thanks for your comment. In the revised manuscript, we have followed your suggestion and provided the quantification of the percentage in the figures.

- **Revised Supplementary Fig. 7c**

[Editorial Note: Figure redacted]

- **Revised Supplementary Fig. 7e**

[Editorial Note: Figure redacted]

- **Revised Supplementary Fig. 7f**

[Editorial Note: Figure redacted]

- **Revised Fig. 3e**

[Editorial Note: Figure redacted]

- **Revised Fig. 5f**

[Editorial Note: Figure redacted]

- **Revised Fig. 7d–f**

[Editorial Note: Figure redacted]

- **Revised Fig. 7i–k**

[Editorial Note: Figure redacted]

- 4 In figure 7D, the decrease of OSM in SAA+2-DG condition compared to SAA condition is not significant (the contrary is stated in the text page 13). Please explain. Also, because it is the only experiment monitoring glycolytic involvement in OSM secretion, could the authors make a stronger demonstration involving glycolysis in OSM secretion?

Response: Thanks for the comment. We apologize for the incorrect illustration in the original manuscript regarding the statement, “*In SAA-treated neutrophils, the production of OSM was significantly upregulated, and such overproduction can be abrogated by the inhibition of glycolysis with 2DG (Fig. 7D).*” In the original experiment, the *P* value between the two groups was 0.0931, indicating no significant statistical difference in the decrease of OSM in the SAA + 2-DG condition compared to the SAA condition. However, we did observe an obvious decreasing trend. To address the potential bias resulting from the small sample size, we included four additional samples under the same experiment conditions for further analysis. As a result, we found that OSM in SAA + 2-DG condition significantly decreased compared to the SAA condition, with a *P* value was 0.0007 (**revised Fig. 5d**).

Additionally, to investigate the involvement of glycolysis in OSM secretion, we conducted experiments using a LDHa inhibitor, FX-11, on SAA-treated neutrophils. The supernatant levels of OSM were measured using an ELISA kit (**revised methods: page 21, line 504–508**). As the results showed that treatment with FX-11 actually increased the secretion of OSM, rather than decreasing it. Thus, glycolysis may regulate OSM

secretion in a distinct mechanism differ from its regulation on PD-L1 expression, and further study is warranted to delineate the detailed mechanism.

As 2-DG and FX-11 targeting different key enzymes of hexokinase-2 (HK2) and LDHA of glycolysis, it is likely that glycolysis may regulate OSM secretion in a distinct mechanism differ from its regulation on PD-L1 expression. For example, Inhibition of LDHA by FX-11 leads to increased pyruvate synthesis, subsequently enhancing the production of acetyl-CoA and promoting cellular tricarboxylic acid (TCA) cycle metabolism, which may upregulate the NF- κ B pathway, thereby promoting OSM secretion (1,2). Additionally, FX-11 has been demonstrated to stimulate the generation of reactive oxygen species (ROS) within cells, and increased ROS levels have also been associated with enhanced OSM secretion (3). Therefore, it is likely these effects of LDHA inhibitor FX-11 different from 2-DG may explain the different effects on OSM secretion in SAA+ neutrophils. Further studies are warranted to delineate the detailed mechanism by which glycolysis regulates OSM secretion.

- **Modified Fig. 5d**

[Editorial Note: Figure redacted]

Figure 5. SAA upregulates OSM production in neutrophils and reduces IFN- γ and TNF- α release in autologous CD8+ T cells. (d) The level of OSM and ARG1 is determined in SAA-neutrophils exposed to glycolysis inhibitor-2DG (25 mmol/L for 12 h) by ELISA (n = 10).

- **Revised Supplementary Fig. 13**

[Editorial Note: Figure redacted]

Supplementary Fig. 13. The relationship between OSM and LDHA inhibitor (FX-11). (a-b) The supernatant level of OSM (a) and ARG1 (b) after purified neutrophils stimulated with DMSO, SAA (1 μ g/mL), LDHA inhibitor-FX-11 (50 μ mol/L), SAA (1 μ g/mL) plus FX-11 (50 μ mol/L), OSM inhibitor-OSM-SMI-10B (50 μ mol/L) or SAA (1 μ g/mL) plus OSM-SMI-10B (50 μ mol/L) were examined by ELISA kits. The addition of FX-11 could elevate the supernatant level of OSM and ARG1 in SAA-treated neutrophils.

References:

1. Chang H, Xu Q, Li J, et al. Lactate secreted by PKM2 upregulation promotes Galectin-

9-mediated immunosuppression via inhibiting NF- κ B pathway in HNSCC. *Cell Death Dis* 2021;12:725.

2. Wang X, Liu R, Qu X, et al. α -Ketoglutarate-Activated NF- κ B Signaling Promotes Compensatory Glucose Uptake and Brain Tumor Development. *Mol Cell* 2019;76:148-162.

3. Le A, Cooper CR, Gouw AM, et al. Inhibition of lactate dehydrogenase A induces oxidative stress and inhibits tumor progression. *Proc Natl Acad Sci U S A* 2010;107:2037-2042.

5 It is important to show the link between OSM secretion by SAA-stimulated neutrophils and CD8 T cell activation. Could the authors make an experiment in which neutrophils are treated with SAA or are treated with SAA and several doses of an OSM inhibitor, and monitor CD8 T cell activation (by flow cytometry as in Fig. 7F)?

Response: Thanks for your very important suggestion. Following your suggestion, we conducted a coculture experiment to validate the relationship between OSM and CD8+ T cells. Pre-activated CD8+ T cells were cocultured with neutrophils for 12h and, subsequently, they were subjected for FCS analysis (**revised methods: page 20, line 596 to page 21, line 503**). The results revealed that the activation of CD8+ T cells was inhibited when cocultured with neutrophils treated with SAA. However, this inhibition was reversed when an OSM inhibitor was added to the coculture system. These findings suggest that OSM plays a role in modulating the activation of CD8+ T cells by SAA-treated neutrophils (**revised results: page 11, line 245–249**).

● **Revised Fig. 5f**

[**Editorial Note:** Figure redacted]

Fig. 5. SAA upregulates OSM production in neutrophils and reduces IFN- γ and TNF- α release in autologous CD8+ T cells. (f) Neutrophils treated with medium, SAA (1 μ g/mL), OSM inhibitor (50 μ M) and SAA (1 μ g/mL) plus OSM inhibitor (50 μ M) for 12h are used to co-culture with autologous CD8+ T cells. The levels of IFN- γ and TNF- α production in CD8+ T cells were analyzed after 12 hours of co-culture by flow cytometry (n = 3).

Minor

- 1 in vitro experiments have been performed with blood-derived neutrophils from HCC patients. If taken from healthy donor, do neutrophils also increase expression of PDL1, glycolytic genes and become more glycolytic in response to SAA?

Response: Thanks for your comment. Per your suggestion, we stimulated neutrophils isolated from the peripheral blood of healthy donors with SAA. Subsequently, we performed qRT-PCR to analyzed the transcription levels of PD-L1 and glycolysis enzymes, including *GLUT1*, *PKM2*, *PFKFB2*, *PFKFB3*, *LDHA*, *GAPDH*, *PFKL*, *ALDOA*, and *ALDOC*. The results showed that, similar to HCC patients, the expression of *PD-L1* and most glycolytic genes increased in SAA-treated neutrophils derived from the peripheral blood of healthy donors. Particularly, there was a significant increase observed in the expression of *PFKFB3* (**revised results: page 9, line 210–212**).

- **Revised Supplementary Fig. 11f**

[Editorial Note: Figure redacted]

Supplementary Fig. 11. SAA mediates glycolytic activation on neutrophils. (f) Neutrophils purified from peripheral blood healthy donors (n = 3) were treated with SAA (1 µg/mL) for 0.5, 6, and 12 hours. The levels of glycolysis-related gene expression were quantified by qPCR. The mRNA expression of PD-L1 and glycolytic enzymes including *PFKFB3*, *LDHA*, *PFKL*, *ALDOA*, and *ALDOC* were significantly increased in SAA-treated neutrophils.

- 2 Is SAA-mediated OSM upregulation also inhibited upon LDHA blockade?

Response: Thanks for your comment. In the response to the 4th major comment, we conducted experiments using a LDHa inhibitor, FX-11, on neutrophils to investigate the involvement of glycolysis in OSM secretion. The supernatant levels of OSM were measured using an ELISA kit. As the results showed that treatment with FX-11 actually

increased the secretion of OSM, rather than decreasing it. Additionally, FX-11 treatment also led to an increase in the production of ARG1 (**revised Supplementary Fig. 13**). Based on these results, we reasoned that LDHa may have a distinct mechanism compared to 2-DG in regulating OSM secretion. It is possible that LDHa inhibition redirects pyruvate towards the generation of acetyl-CoA for entry into the tricarboxylic acid (TCA) cycle. This metabolic shift could explain why inhibiting LDHa does not result in decreased OSM secretion. We would like to address the detailed mechanism by which glycolysis mediates SAA-induced OSM upregulation in the future study.

3 Could authors choose images in Fig. 3A that are representative of their quantifications (Fig. 3B)? Visually, the important infiltration of CD8 T cells in SAA low samples is confusing with quantifications (showing an increase of CD8 T cells infiltration in SAA high samples).

Response: We apologize for the wrong order of quantifications between the SAA low group and high group presented in the original Fig. 3B. We have made the necessary corrections, and now the quantifications of CD8+ T cells in the revised **Fig. 2b** have been correctly ordered.

Consequently, the images in the revised **Fig. 2a** now accurately represent their corresponding quantifications in **Fig. 2b**. Therefore, the statement in the original manuscript ("*neutrophils (CD15+) (P<0.0001), macrophages (CD68+) (P<0.01), and CD8+ T cells (P<0.05) were significantly increased in the SAA-high cohort*") related to CD8+ T cells was also incorrect. In fact, CD8+ T cells were significantly decreased in the SAA-high cohort. We have made the necessary revisions in the manuscript to reflect this correction (**revised results: page 7, line 151–155**). The corresponding figures were shown as follows:

- **Revised Fig 2a & b**

[Editorial Note: Figure redacted]

Figure 2. Association between local SAA and immune cell infiltration in HCC. (a–b) mIHC images show the simultaneous detection of DAPI (blue), SAA (red), CD15+ neutrophils (yellow), CD68+ macrophages (brilliant blue), CD8+ T cells (white), and PD-L1 (green) in human HCC peritumor TMA. The count of infiltrating immune cells between SAA high and SAA low samples is compared using an unpaired *t*-test (n = 160).

4 Could authors check for the presence of glycolytic+PD-L1+CD15+pSTAT3+ cells in peritumour/tumor? And their physical proximity with CD8 T cells?

Response: Thanks for the comment. We utilized LDHA antibody as a glycolytic marker for six-plex IHC analysis. Our findings revealed the presence of LDHA+ PD-L1+ pSTAT3+ CD15+ cells in the peritumor region. Besides, through cellular spatial relation map analysis, we observed a spatial correlation between CD8+ T cells and LDHA+ PD-L1+ pSTAT3+ CD15+ cells in the peritumoral region of HCC tissues (**revised Supplementary Fig. 12e**).

- **Revised Supplementary Fig. 12e**

[Editorial Note: Figure redacted]

Supplementary Fig. 12. Activation of the LDHA/STAT3 rather HIF-1 α /PKM2 axis is involved in the glycolysis-mediated PD-L1 upregulation in SAA-associated neutrophils. (e) The six-plex mIHC assay shows the co-expression of LDHA (yellow), p-STAT3 (red), and PD-L1 (green) in CD15+ (blue) neutrophils from HCC peritumoral specimens, and located close to CD8+ T cells in the HCC peritumoral area.

5 In the context of Stat3 deficiency, the authors stated that IFN γ + cells are T cells. Could authors add a CD3 or CD8 marker to the mIHC panel? (Fig. 9I).

Response: Thanks for the comment. As you suggested, we have added CD8 marker to the mIHC panel in the revised **Fig. 7e and Fig. 7j**.

- **Revised Fig. 7e**

[**Editorial Note:** Figure redacted]

- **Revised Fig. 7j**

[**Editorial Note:** Figure redacted]

6 For the flow cytometry data, can the authors indicate the percentage of what they refer to on the y axis?

Response: Thanks for your comment. We apologize for the unclear indication of the percentage on the y axis in each figure. We have made the necessary revisions and added the appropriate labels indicating the percentage of what is being referred to on the y axis in each figure.

REVIEWER COMMENTS

Reviewer #1 noted to editors that all comments were addressed.

Reviewer #1 considered comments from reviewer #2 addressed.

Reviewer #3 (Remarks to the Author):

The authors have addressed most of my concerns and I thank them for that.

However, a major and a minor issue remain.

Major (Q2): Each of 2-DG and LDHa inhibitor (FX-11) diminished SAA-induced PDL1 expression (Fig. 4f, h), but only 2-DG inhibited SAA-induced OSM secretion, the LDHA inhibitor augmenting it (Fig. 5d vs. Fig. S13). The explanations provided in the rebuttal for additional effects FX-11 could have are unfortunately not convincing, and add confusion as to why the authors chose this inhibitor.

I would suggest the authors either to repeat these experiments with other ways to inhibit glycolysis, or to remove the part linking glycolysis to OSM secretion.

Consequently, the authors cannot state in the abstract that “In vitro experiments demonstrated that SAA induces neutrophils to express PD-L1 and release oncostatin M through glycolytic activation via LDHA/STAT3 pathway”, while LDHA inhibition did not reduce, but instead increased OSM secretion.

For the same point, the reviewer does not understand the meaning of the title of Fig. S13: “The relationship between OSM and LDHA inhibitor (FX-11).”

Minor (Q5):

I thank the authors for having added a CD8 marker to their multiplex-IF, which I had asked to show that IFNG⁺ cells are indeed T cells. However, in their IFs, IFNG⁺ cells do not seem to co-localize with CD8⁺ cells (Fig. 7e) and IFNG⁺ cells seem to co-localize with MPO⁺ cells rather than CD8⁺ cells (Fig. 7j). The authors should comment about this.

Reviewer #3' comments:

Major

Each of 2-DG and LDHa inhibitor (FX-11) diminished SAA-induced PDL1 expression (Fig. 4f, h), but only 2-DG inhibited SAA-induced OSM secretion, the LDHA inhibitor augmenting it (Fig. 5d vs. Fig. S13). The explanations provided in the rebuttal for additional effects FX-11 could have unfortunately not convincing, and add confusion as of why the authors chose this inhibitor. I would suggest the authors either to repeat these experiments with other ways to inhibit glycolysis, or to remove the part linking glycolysis to OSM secretion.

Consequently, the authors cannot state in the abstract that “In vitro experiments demonstrated that SAA induces neutrophils to express PD-L1 and release oncostatin M through glycolytic activation via LDHA/STAT3 pathway”, while LDHA inhibition did not reduce, but instead increased OSM secretion.

For the same point, the reviewer does not understand the meaning of the title of Fig. S13: “The relationship between OSM and LDHA inhibitor (FX-11).”

Response: Thank you for your detailed and constructive comment of our manuscript, particularly regarding the effects of 2-DG and LDHa inhibitor (FX-11) on SAA-induced OSM secretion. Your insights have prompted a rigorous re-evaluation of our findings and methodology.

Regarding your concerns about the differential effects of 2-DG and FX-11, and the rationale behind selecting FX-11 as an inhibitor, we acknowledge that our initial explanations might not have been sufficiently convincing. We appreciate your pointing out the lack of clarity in our rationale for the choice of FX-11 and the related result interpretation.

In light of your feedback, we have taken the following actions:

1. **Specific Re-evaluation of FX-11 Data:** We have carefully reexamined our data concerning FX-11, particularly its augmenting effect on OSM secretion, as contrasted with the inhibitory effect of 2-DG. We agree that our initial interpretation did not adequately address this discrepancy.
2. **Removal of Disputed Content:** In line with your suggestion, we have decided to remove the sections discussing the link between glycolysis, as modulated by 2-DG and FX-11, and OSM secretion. This revision is based on the current ambiguity

surrounding the differential effects of these inhibitors and the need for further investigation to provide a more conclusive understanding.

3. **Modification of Figures and Text:** We have updated Fig. 5d and removed Supplementary Fig. 13 in line with the exclusion of the related content. The text in the results section has been appropriately modified to align with these adjustments (page 10, line 25 to 26).

Revised Fig. 5d:

[Editorial Note: Figure redacted]

4. **Revised Abstract:** The abstract has been revised to align with the revised content of the manuscript, focusing on the more robust and conclusive aspects of our study.

Revised abstract:

“In vitro experiments demonstrated that SAA induced neutrophils to express PD-L1 through glycolytic activation via LDHA/STAT3 pathway and to release oncostatin M, thereby attenuating cytotoxic T cell activities.”

Minor

I thank the authors for having added a CD8 marker to their multiplex-IF, which I had asked to show that IFNG⁺ cells are indeed T cells. However, in their IFs, IFNG⁺ cells do not seem to co-localize with CD8⁺ cells (Fig. 7e) and IFNG⁺ cells seem to co-localize with MPO⁺ cells rather than CD8⁺ cells (Fig. 7j). The authors should comment about this.

Response: Thank you for your insightful feedback regarding the co-localization of IFN γ ⁺ cells with CD8⁺ T cells in the multiplex-IF analysis, as depicted in our original Fig. 7e and 7j. In response to your observation, we meticulously re-evaluated our multiplex-IF images and discovered an error in the fusion of channels. We sincerely apologize for the negligence in accurately describing the CD8⁺ T cells and IFN γ ⁺ cells in the original figures. We have now corrected this technical error, and the revised Fig. 7e and Fig. 7j accurately exhibit the co-localization of CD8⁺ T cells and IFN γ ⁺ cells.

- **Revised Fig. 7e**

[Editorial Note: Figure redacted]

- **Revised Fig. 7i**

[Editorial Note: Figure redacted]

REVIEWERS' COMMENTS

Reviewer #3 (Remarks to the Author):

The authors have now addressed all my previous comments.

Reviewer: Etienne Meylan